# VIDEO2POLICY: SCALING UP MANIPULATION TASKS IN SIMULATION THROUGH INTERNET VIDEOS

## ABSTRACT

Simulation offers a promising approach for cheaply scaling training data for generalist policies. To scalably generate data from diverse and realistic tasks, existing algorithms either rely on large language models (LLMs) that may hallucinate tasks not interesting for robotics; or digital twins, which require careful real-to-sim alignment and are hard to scale. To address these challenges, we introduce Video2Policy, a novel framework that leverages large amounts of internet RGB videos to reconstruct tasks based on everyday human behavior. Our approach comprises two phases: (1) task generation through object mesh reconstruction and 6D position tracking; and (2) reinforcement learning utilizing LLM-generated reward functions and iterative in-context reward reflection for the task. We demonstrate the efficacy of Video2Policy by reconstructing over 100 videos from the Something-Something-v2 (SSv2) dataset, which depicts diverse and complex human behaviors on 9 different tasks. Our method can successfully train RL policies on such tasks, including complex and challenging tasks such as throwing. Furthermore, we show that a generalist policy trained on the collected sim data generalizes effectively to new tasks and outperforms prior approaches. Finally, we show the performance of our policies improves by simply including more internet videos. We believe that the proposed Video2Policy framework is a step towards generalist policies that can execute practical robotic tasks based on everyday human behavior.

## 1 INTRODUCTION

Training generalist policies requires collecting large quantities of diverse robotic expert data. However, the approach of collecting data through teleoperation is constrained by high operation costs, while collecting data from autonomous policies can be unsafe, or result in low-quality data. Simulation offers an appealing alternative to real-world data that does not suffer from these challenges, and can be used to train general and robust policies (Hwangbo et al., 2019; Andrychowicz et al., 2020). Recent work has explored automatically generating diverse and relevant tasks in simulation as a way to create a scalable pipeline for generating robotics data (Deitke et al., 2022; Wang et al., 2023b;c; Makatura et al., 2023).

However, existing methods primarily rely on text-only task specification using Large Language Models (LLMs), which do not have grounded robotics knowledge. They often produce tasks that are not diverse, have uninteresting behavior, or use uninteresting object assets and are thus less useful for training generalist policies. To better capture the real-world distributions of task behaviors and objects, we propose leveraging RGB videos from the internet to create corresponding tasks. Unlike Real2Sim approaches that construct digital twins (Hsu et al., 2023; Torne et al., 2024) for a single scene, we want to train a generalist policy for multiple scenes and therefore we do not require perfect reconstructions. Instead, we leverage large amounts of internet videos to capture task-relevant information such as object assets and scene layouts. We then generate simulated tasks using a Vision-Language Model (VLM) that can take the video, video captions, object meshes, sizes, and 6D poses, and produce corresponding **task codes**, which can be executed to generate scenes.

Beyond task proposals, we require an efficient and automatic way to solve tasks. Naively applying reinforcement or imitation learning is challenging as it requires manual human effort for each task to create demonstrations or reward functions. Inspired by the recent advancements of LLMs in

Figure 1: The **Video2Policy** framework can leverage internet videos to generate simulation tasks and learn policies for them automatically, which can be considered a data engine for generalist policies.

code generation for various tasks (Achiam et al., 2023; Roziere et al., 2023), some researchers have proposed automating policy learning or deployment through using an LLM to produce policy code directly (Huang et al., 2023b; Liang et al., 2023; Wang et al., 2023b), or to produce reward function code (Ma et al., 2023; Wang et al., 2023c). Gensim (Wang et al., 2023b) leverages this idea for unsupervised task generation. However, predicting goals restricts it to simple tasks as it does not account for dynamic tasks or tasks that involve complex object interactions.

In contrast, reinforcement learning (RL) is effective at solving complex tasks (Schulman et al., 2017; Ye et al., 2021; Hafner et al., 2023; Wang et al., 2024; Springenberg et al., 2024). RoboGen (Wang et al., 2023c) leverages LLM-generated reward functions for RL. However, it is hard to scale as it requires manual success functions. However, since we leverage both text prompts and explicit visual prior knowledge from human videos, we can leverage that information for better success functions.

We propose **Video2Policy**, a framework to leverage internet RGB videos in an automated pipeline to create simulated data to train generalist police, as shown in Fig. 1. It produces code for task generation in simulation using VLMs and learns policies for those tasks via reinforcement learning with reward reflection. This autonomous approach allow us to easily scale up data generation by using internet videos to produce a large amount of visually grounded tasks. Our framework leverages both behavioral and object diversity from the videos, enabling generalization both at the object level as well as task level. To further improve generalization instead of simply cloning the trajectory from the video, we apply position randomization to the initial states. Specifically, our framework consists of two phases: (1) We reconstruct the object meshes involved in the tasks from videos as task assets, and extract the 6D pose of each object (2) We leverage the VLM to write the task code based on visual information as well as text prompts, and iteratively generate the reward function using reward reflection based on training logs and evaluation results. Ultimately, we obtain the corresponding refined task code along with the learned policy model that demonstrates behavior similar to the input video, enabling it to generate high-quality data relevant to daily human tasks.

For experimental evaluation, we focus on table-top manipulation tasks with a single robot arm in IsaacGym simulator(Makoviychuk et al., 2021). To validate the effectiveness of our approach, we conduct experiments on the Something-Something V2 (SSv2) dataset (Goyal et al., 2017), a human video dataset with diverse scenes and language instructions of human daily behaviors. We reconstruct over 100 videos on 9 distinct tasks from the dataset in total, including involving multiple objects. To further evaluate more complex behaviors and object relationships, we also utilize three self-recorded videos. Significantly, the results indicate the policies learned by our method significantly outperform those learned from previous LLM-driven methods. We achieve 82% success rates for the 9 tasks on average. To demonstrate generalization capabilities, we train a general policy by imitation learning from the simulation data collected from the learned policies from 50 videos of the same behavior. We find that it can achieve 75% success rates on the tasks from the 10 unseen videos with the same behavior. More importantly, as the number of training tasks increases, the performance of the general policy improves, which shows the scalability of our framework.

## 2 RELATED WORK

**Real2Sim Scene Generation for Robotics** Generating realistic and diverse scenes for robotics has recently emerged as a significant challenge aimed at addressing data issues through simulation. Some researchers have developed Real2Sim pipelines for image-to-scene or video-to-scene generation. Certain studies (Hsu et al., 2023; Torne et al., 2024) focus on constructing digital twins that facilitate the transition from the real world to simulation; however, these approaches often depend on specific real-world scans or extensive human assistance. Dai et al. introduces the concept of digital cousins, while Chen et al. (2024) employs inverse graphics to enhance data diversity. Nevertheless, their approach to diversity primarily involves replacing various assets. The lack of task-level diversity hinders the ability to capture the distribution of real-world tasks, and the constraints of specific data formats complicate the scalability of robotics data. Although Real2Code (Mandi et al., 2024) aims to build simulation scenes from RGB images, it focuses on articulation parts and requires in-domain code data for fine-tuning.

**Scaling up Simulation Tasks** Previously, researchers aimed to build simulation benchmarks to facilitate scalable skill learning and standardized workflows (Li et al., 2023; Gu et al., 2023; Srivastava et al., 2022; Nasiriany et al., 2024). Most of these benchmarks were constructed manually, making them difficult to scale. Recently, some researchers have focused on text-to-scene generation, emphasizing the creation of diverse scenes. Works such as Deitke et al. (2022); Makatura et al. (2023); Liang et al.; Chen et al. (2023) utilize procedural asset generation or domain randomization, while Jun & Nichol (2023); Yu et al. (2023a); Poole et al. (2022) engage in text-to-3D asset generation. Although these approaches can achieve asset-level or scene-level diversity in robotic tasks, they fall short in delivering task-level diversity. Gensim (Wang et al., 2023b) attempts to generate rich simulation environments and expert demonstrations using large language models (LLMs) to achieve task-level diversity. However, text-based task generation tends to be arbitrary regarding object selection and their relationships, limiting its ability to represent the true distribution of tasks in the real world. In contrast, our work leverages real-world RGB videos to create corresponding simulation tasks that better reflect the real-world distributions of tasks and objects, facilitating easier scalability due to the abundance and accessibility of internet video data.

**Policy Learning via LLMs** To enable automatic policy learning with high quality, researchers are increasingly turning to large language models (LLMs) for assistance. Some studies (Liang et al., 2023; Huang et al., 2023a; Lin et al., 2023; Wang et al., 2023a) propose generating structured code outputs for decision-making problems, most of which rely on predefined primitives. Other works (Yu et al., 2023b; Ma et al., 2023; Wang et al., 2023c) generate reward functions using LLMs for reinforcement learning. Nevertheless, Eureka (Ma et al., 2023) requires predefined success functions for iterative updates of reward functions, while RoboGen (Wang et al., 2023c) selects the highest reward as the initial state for the next sub-task, which introduces noise due to the variability in the generated reward functions. In contrast, our work generates success functions by leveraging visual prior knowledge from the provided videos and updates the reward functions iteratively using a Chain-of-Thought (CoT) approach (Wei et al., 2022).

## 3 BACKGROUND

To present our work systematically, we formulate it as a two-level hierarchical solution of Markov Decision Processes (MDPs), corresponding to the two-phase pipeline. In contrast to some hierarchical methods targeting action levels (McGovern et al., 1998; Hauskrecht et al., 2013), we focus on the level of reward functions.

**Low-level MDP of Controlling Problem** Specifically, the scene reconstruction phase is to construct the MDP $\mathcal{G} = \langle \mathcal{S}, \mathcal{A}, T, \mathcal{R}_{0|1} \rangle$ from videos, where $\mathcal{S} \in \mathbb{R}^m$ represents the states of the environment, $\mathcal{A}$ is the action space of the agent, and $T$ is the transition probability function. $\mathcal{R}_{0|1}$ is the 0-1 reward function that distinguish whether the trajectory is successful. $\mathcal{R}_{0|1}$ can be a scalar evaluation function. To solve this MDP, we will train a policy $\pi$ through reinforcement learning in the Isaac Gym simulation. To achieve high performance, we leverage the LLM to sample various reward functions to learn policies in a high-level manner, based on the evaluation results from $\mathcal{R}_{0|1}$ and some CoT (Wei et al., 2022) instructions.

**High-level MDP of Reward Designs** The aim of reward design is to create a shaped reward function that simplifies the optimization of a challenging given reward function, such as the sparse 0-1 reward function $\mathcal{R}_{0|1}$. Following the definition of Reward Design Problem (RDP) from previous works (Singh et al., 2009; Ma et al., 2023), we consider a high-level MDP $\hat{\mathcal{G}} = \langle \hat{\mathcal{S}}, \hat{\mathcal{A}}, \hat{T}, F \rangle$. Here, $\hat{\mathcal{A}}$ is the space of reward functions. Each time we choose an action $\hat{R} \in \mathcal{A}$ in the MDP $\hat{\mathcal{G}}$, we will train a policy $\pi$ by RL for the low-level MDP $\mathcal{G} = \langle \mathcal{S}, \mathcal{A}, T, \hat{\mathcal{R}} + \mathcal{R}_{0|1} \rangle$. The horizon of the MDP $\hat{\mathcal{G}}$ is the iteration number in the second phase. $\hat{\mathcal{S}}$ includes the training and evaluation information during RL and the policy model $\pi$. $\hat{T}$ is the state transition function, and $F$ is the reward function that produces a scalar evaluation of any policy $\pi$. Specifically, $F$ is equal to $\mathcal{R}_{0|1}$. Thus, the high-level MDP's goal is to find a reward function $\hat{\mathcal{R}} \in \hat{A}$ to maximize the success rates of the low-level policies.

# 4 GENERATING SIMULATED TASKS AND POLICIES FROM HUMAN VIDEOS

The proposed framework, **Video2Policy**, steps further for task proposal and policy learning through internet videos, to provide diverse and realistic tasks as well as the corresponding learned policies. It consists of two phases: task scene generation and policy learning. In Sec. 4.1, we introduce the pipeline for reconstructing scenes from RGB videos. Subsequently, in Sec. 4.2, we demonstrate how to generate the code for the task and learn policies to solve it. Finally, we provide an example of training a generalist policy within our framework in Sec. 4.3.

## 4.1 SCENE RECONSTRUCTION FROM VIDEOS

Since the goal of this work is to learn policies from videos rather than automatically retarget trajectories, our scene reconstruction phase focuses on reconstructing the manipulated objects along with their relative relationships. To master the skill demonstrated in the video, we allow for random positions and orientations of each object in the initial states. Consequently, the pipeline for the scene reconstruction is illustrated in Fig. 2: (1) detect and segment the manipulated objects in the video using text captions; (2) reconstruct the object meshes from the video and estimate the actual sizes of the meshes; (3) perform 6D position tracking for each object in the video. Afterward, we obtain a JSON file that includes the video and object information, from which the task code is generated.

**Object Grounding** We first use Grounding DINO (Liu et al., 2023) to detect the manipulated objects in the first frame of the video based on their names. Since the SSv2 dataset (Goyal et al., 2017) provides both video captions and object labels, we use these as text prompts for detection. For self-collected videos of more challenging behaviors, we provide the video captions and object names manually. Afterward, we perform video segmentation using the SAM-2 model (Ravi et al., 2024). Specifically, we segment the objects in the first frame using bounding boxes obtained during detection and select five positive pixel points in each object mask for video segmentation. This process yields segmented videos that contain the masks of the manipulated objects.

**Object Reconstruction** With the segmentation masks of each object for each frame, we perform mesh reconstruction based on these images. Since most internet videos are recorded from a single viewpoint, we leverage the InstantMesh model (Xu et al., 2024) to reconstruct the 3D meshes, which supports mesh generation from a single image. Typically, we choose the first frame to reconstruct the meshes; however, for those with objects that are significantly occluded in the first frame, we utilize the last frame instead. Due to the training mechanism in the current mesh generation methods, the meshes are commonly normalized during reconstruction, leading to mismatched sizes compared to the corresponding objects in the video. To establish a more realistic size relationship between objects, we propose a simple and efficient size estimation method. We predict the camera intrinsic matrix $\mathbb{K}$ and the depth $d_{i,j}$ of the image $I_{i,j}$ using UniDepth (Piccinelli et al., 2024), where $i, j$ are the pixel coordinates. Given the masked region $M$ of the object, we can calculate the maximum distance for the masked region in reality $D_{\text{image}}$: $D_{\text{image}} = \max_{(i_1,j_1),(i_2,j_2)\in M} \|\mathbf{p}(i_1, j_1) - \mathbf{p}(i_2, j_2)\|$, where $\mathbf{p}(i,j) = \mathbb{K}^{-1} \cdot [x, y, 1]^{\text{T}} \cdot d_{i,j}$. Here $\mathbf{p}$ is the 3D position of each masked pixel in the camera coordinate system. We can then calculate the maximum distance of vertices in the mesh object, denoted as $D_{\text{mesh}}$. The scale ratio $\rho$ for the mesh object is defined as $\rho = D_{\text{image}}/D_{\text{mesh}}$. The absolute sizes may exhibit some noise due to errors in depth estimation, intrinsic estimation, and object occlusion. However, the relative size of each object is mostly accurate, as $D_{\text{image}}$ and $D_{\text{image}}$ are calculated within the same camera coordinate

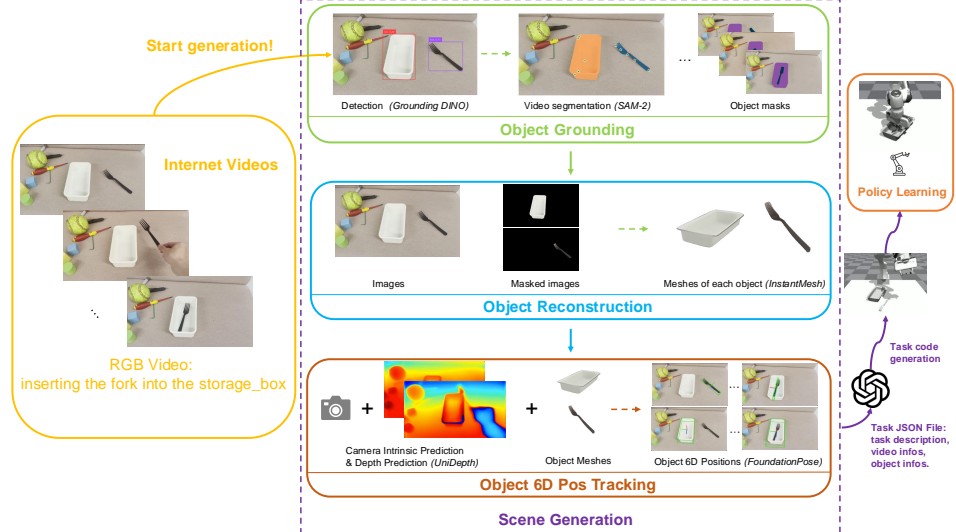

Figure 2: **The scene generation phase of the proposed Video2Policy framework**. We generate the tasks from internet videos into the simulation, which we attempt to solve by reinforcement learning.

system. In practice, we can apply domain randomization to change the absolute sizes of each object while maintaining the same relative size among them.

**6D Position Tracking of Objects** After reconstructing the objects, we predict the 6D position of each object throughout the video, which will be fed into GPT-4o for code generation. We utilize FoundationPose (Wen et al., 2024) in model-based setups to estimate the position and orientation of the objects. This model takes the object mesh, predicted camera intrinsics, and the depth information from each frame as inputs. Finally, we automatically generate a URDF file for each object based on the mesh file and the calculated scaled size. Ablations for the tracking information are in App. A.3.

Compared to directly feeding the videos into the task generation process, we explicitly leverage visual prior knowledge from the videos and provide comprehensive information about the task, including the 3D meshes, object sizes, and 6D positions of the objects. We consolidate all the information into a JSON file that describes the task scene. Consequently, the generated tasks can better align with the distributions of real-world scenes and behaviors.

## 4.2 TASK CODE GENERATION AND POLICY LEARNING

After extracting the visual information from the video into a task JSON file, we can build the task scene in simulation and learn policies based on GPT-4o. This process occurs in two stages. First, we generate the complete task code, which can be executed directly in the Isaac Gym (Makoviychuk et al., 2021) environment. Second, inspired by recent work on LLM-driven reward function generation, we iteratively fine-tune the generated reward function using in-context reward reflection (Shinn et al., 2023; Ma et al., 2023; Wang et al., 2023a). In contrast to the previous work Eureka (Ma et al., 2023), which is the most similar to ours, we generate the task codes, including the reward functions, from scratch, rather than relying on pre-existing task codes and starting reward reflection from manually defined success functions. We formulate this learning process as a two-level hierarchical solution of MDP, defined in Sec. 3.

**Task Code Generation** Inspired by previous work (Ma et al., 2023; Wang et al., 2023b;c), we systematically introduce the pipeline for general task code generation, which helps to infer codes by prompting in a well-organized way. Notably, the task code consists of six parts: scene information, reset function, success function, observation function, observation space function, and reward function. (1) The scene information refers to the task scene JSON file created from the videos. It contains the task title, video file path, video description, and object information, including sizes, URDF paths, and tracked 6D position lists. (2) The reset function is responsible for positioning the objects according to specific spatial relationships in the beginning. For instance, in the task "uncovering A from B," both objects A and B are randomly placed on the table by the parent reset function, but object A will be positioned above object B as specified by the generated child reset function. (3) The success function determines the success state. Notably, both the reset and success functions

are generated by GPT-4o based on the task description, the provided 6D position list, and Chain-of-Thoughts examples (Wei et al., 2022). (4) Furthermore, since we use reinforcement learning to master the skill in simulation, we have access to the states of the objects, and the policy is based on state observations. Thus, in addition to the object states, we also query GPT-4o to determine whether additional observations are necessary. Interestingly, we find that it can include observations such as the distance between the object and the gripper or the normalized velocity toward the target object. (5) Simultaneously, it calculates the correct observation shape for building the neural networks. (6) Regarding the reward function, we follow the instructions from Ma et al. (2023) with CoT examples. We write a template for task code generation, allowing us to query the VLM just once to generate the executable code encompassing all six parts. Most significantly, we generate eight example codes and select one by re-checking the correctness, reasonability and efficiency from GPT-4o, which is the base code candidate for the subsequent in-context reward reflection stage. The generated examples are demonstrated in App. A.1 and the robustness evaluation results are in App. A.2.

**Reinforcement Learning and Reward Function Iteration** As mentioned in 3, we propose to optimize the policy through a two-level hierarchical solution of MDP. Given the generated task code, we train policies through reinforcement learning under the reward function $\hat{\mathcal{R}}$ and success function $\mathcal{R}_{0|1}$. Notably, we assign a high trade-off to the success rewards, formulating the training reward function as $\hat{\mathcal{R}} + \lambda\mathcal{R}_{0|1}, \lambda = 100$. Moreover, following the approach in Eureka (Ma et al., 2023), we apply the in-context reward reflection to design the reward functions iteratively using GPT-4o. Each time, we sample $N = 8$ different reward functions for training policies and collect the training and evaluation logs. We then select the best function from the previous iteration and generate new reward functions based on these logs, along with specific instructions and CoT examples. For example, in addition to providing good examples from previous tasks, we prioritize training outputs where the accumulated rewards for successful trajectories exceed those for failed ones.

### 4.3 TRAINING GENERAL POLICIES IN SIMULATION

As witnessed by the recent success of policy learning from large-scale datasets with certain formats (Padalkar et al., 2023; Reed et al., 2022; Team et al., 2024; Brohan et al., 2023), we want to investigate how to learn a general policy from the internet videos, which directly outputs the executable actions of the robot rather than the un-executable future videos (Du et al., 2024; Qin et al., 2023) or language tokens (Liang et al., 2023; Brohan et al., 2023). We consider our Video2Policy a data engine to generate successful policies from internet videos. Then we can acquire expert trajectories in simulation, which match the video behavior. Notably, those expert trajectories can be any format we want, such as states, 2D images, or 3D images. In this work, we choose RGB image observation as the universal perception format of our general policy. Therefore, we train policies from the videos and collect the successful trajectories from the learned policy. Afterward, we use imitation learning to learn the general policy from the collected dataset by behavior cloning.

## 5 EXPERIMENTS

In this section, we present detailed evaluations of the proposed Video2Policy framework on internet video datasets about manipulations. Specifically, the experiments are designed to answer the following questions: (**a**) How does the generated scene from the video look like, including the objects as well as the visual information, in Sec. 5.1. **b)** How does the policy trained under our framework perform compared to the videos and the baselines, in Sec. 5.1. (**c**) What is the performance of the general policy learned from diverse internet videos, and can it generalize to novel scenes, in Sec. 5.2. (**d**) What affects the proposed Video2Policy framework most for policy learning, in Sec. 5.3.

**Experimental Setup** We use the Issac Gym (Makoviychuk et al., 2021) as the simulation engine for all the experiments, which is commonly used in robotics tasks due to the advantages of efficient computing and highly realistic physics. We focus on the table manipulation tasks, and all the objects will randomly reset on the table in the beginning. The horizon of all the tasks is set to 300 and the parallel environments are 8192, equally. For each task, we average the success rates over 10 evaluation episodes across 3 runs with different seeds.

**Video Data Source** To reconstruct scenes from internet RGB videos, we choose the Something Something V2 (SSv2) dataset (Goyal et al., 2017), a common and diverse video dataset for the

Figure 3: Some visualization of the tasks generated from SSv2 Video Dataset.

robotics community. It includes diverse behaviors concerning manipulating something with something by human hands. To further investigate the ability of our framework on more complex objects or behaviors, we record three in-the-wild videos of different behaviors by ourselves. Notably, all the videos we use in the experiment are 3 channels with RGB only, with the highest accessibility. For the video quality, the small motion of the camera is tolerated, and we scale the resolution to 1024.

**Scene Generation** As mentioned in Sec. 4.1, we do 6D position tracking for all the objects. Considering that we randomize the initial states of all the objects, we only feed the 6D pos from the first frame and the final frame into the prompts. It is reasonable in most tasks because those two frames are significant to infer the relationship among objects. Even if this simplification will miss the motion information for some behavior, e.g. throwing, we also provide the task description to design the reward function so that the LLM will generate the velocity reward components. Moreover, we explicitly calculate the difference between the 6D pos and feed the information into the LLM to think about the success function. For most of the SSv2 videos of a single object, there are severe occlusions, making it difficult to reconstruct the mesh asset. We manually choose the first frame or the last frame to reconstruct the mesh and predict the 6D position in the same pipeline. We generate the task code in a curriculum manner (Ma et al., 2023; Wang et al., 2023b) after obtaining the visual information. From the beginning, we provide the example code of `reach a block` and `grasp a block`. Then we will add the successfully generated task examples into the task pool for the next one. Finally, it can even learn to use the direction velocity reward for dynamic tasks and resolve them. Some demos for the generated tasks are in Fig. 3.

**Reinforcement Learning** For the policy learning, we choose the PPO (Schulman et al., 2017) algorithm in a well-tuned implementation codebase (Makoviichuk & Makoviychuk, 2021; Ma et al., 2023). We share the same parameters recommended in the codebase across all tasks and all baseliens. As for the evaluation metric, we write the ground-truth success function for each generated task. Unlike Eureka (Ma et al., 2023), we do not allow access to the evaluation metric during training, and we manually evaluate the results for the final models. For SSv2 dataset, we make 5 iterations and sample 8 reward functions at each iteration during reinforcement learning. In our collected videos, we make 8 iterations and sample 8 reward functions.

## 5.1 POLICY LEARNING FROM VIDEOS

**Policy Learning Baselines** Since there is few works transforming the internet RGB videos into policies for the task, we focus on the policy learning part in the experiments. We benchmark our method against the following baselines. (1) Code-as-Policy (CoP) (Liang et al., 2023), which queries the LLM with all the states in the environment to write the executable code for the robot. To ensure better performance of CoP, we use the close-loop control and regenerate code policies every 50 steps. (2) RoboGen, which does not require a success function and learns without reward reflection iteration. (3) Eureka, which generates code for both the reward and the success function using an LLM and does not use video information. To make fair comparisons, we use the same object meshes and task codes generated from the videos for all the baselines.

**Performance Analysis of Learned Policies** We compare the performance of our method with the above baselines on 9 videos sampled from 9 tasks generated from SSv2 dataset, as shown in Tab. 1. We find that the proposed Video2Policy method outperforms the baseline in most tasks, with smaller variance across seeds. RoboGen (Wang et al., 2023c) and Eureka (Ma et al., 2023) achieve comparable results to ours in the videos of a single object. However, for multiple objects, the

Table 1: **Results of Learned Policies for Videos in SSv2 dataset (3 seeds)**. The mean $\pm$ std of the success rates are shown in the table. Our method outperforms the other baselines to a degree and achieves smaller variance in general.

| Task (Succ.) | Code-as-Policy | RoboGen | Eureka | Video2Policy |
|---|---|---|---|---|
| *single object* | | | | |
| **Push sth. left** | $0.17 \pm 0.13$ | $0.93 \pm 0.05$ | $\mathbf{1.00} \pm 0.00$ | $\mathbf{1.00} \pm 0.00$ |
| **Push sth. right** | $0.75 \pm 0.12$ | $\mathbf{1.00} \pm 0.00$ | $\mathbf{1.00} \pm 0.00$ | $\mathbf{1.00} \pm 0.00$ |
| **Lift up sth.** | $0.33 \pm 0.21$ | $0.28 \pm 0.09$ | $0.83 \pm 0.13$ | $\mathbf{0.93} \pm 0.05$ |
| **Tip sth. over** | $\mathbf{1.00} \pm 0.00$ | $0.97 \pm 0.05$ | $0.67 \pm 0.47$ | $\mathbf{1.00} \pm 0.00$ |
| *multiple objects* | | | | |
| **Cover sth. with sth.** | $0.00 \pm 0.00$ | $0.00 \pm 0.00$ | $0.00 \pm 0.00$ | $\mathbf{0.07} \pm 0.05$ |
| **Uncover sth. from sth.** | $0.10 \pm 0.08$ | $0.67 \pm 0.26$ | $0.63 \pm 0.38$ | $\mathbf{0.97} \pm 0.05$ |
| **Push sth. with sth.** | $0.03 \pm 0.05$ | $0.03 \pm 0.05$ | $\mathbf{0.47} \pm 0.38$ | $0.43 \pm 0.40$ |
| **Push sth. next to sth.** | $0.80 \pm 0.16$ | $0.23 \pm 0.05$ | $0.83 \pm 0.12$ | $\mathbf{1.00} \pm 0.00$ |
| **Drop sth. in front of sth.** | $0.53 \pm 0.31$ | $0.37 \pm 0.09$ | $0.87 \pm 0.17$ | $\mathbf{0.93} \pm 0.05$ |
| **Average** | 0.41 | 0.50 | 0.70 | **0.82** |

Table 2: **Results of Learned Policies for Videos of self-collected videos (3 seeds)**. The mean $\pm$ std of the success rates are shown in the table. For the videos concerning more complex objects or behaviors, such as throwing, we achieved significantly better performance compared to the baselines.

| Task (Succ.) | Code-as-Policy | RoboGen | Eureka | Video2Policy |
|---|---|---|---|---|
| **Insert Fork into Container** | $0.07 \pm 0.05$ | $0.27 \pm 0.38$ | $0.57 \pm 0.40$ | $\mathbf{0.93} \pm 0.05$ |
| **Sliding Remoter to Mouse** | $0.00 \pm 0.00$ | $0.53 \pm 0.12$ | $0.87 \pm 0.05$ | $\mathbf{0.97} \pm 0.05$ |
| **Throw Garlic into Bowl.** | $0.00 \pm 0.00$ | $0.03 \pm 0.05$ | $0.37 \pm 0.29$ | $\mathbf{0.70} \pm 0.36$ |
| **Average** | 0.02 | 0.28 | 0.60 | **0.87** |

performance of RoboGen drops a lot, while the Eureka has much larger variances during training. It indicates that the success function is significant for more complex tasks. Moreover, the visual information of the object relationship is important for proposing the success function.

To further demonstrate the ability of Video2Policy in policy learning, we propose three harder tasks from self-collected videos, including the non-convex object manipulation and dynamic tasks. The results are illustrated on Tab. 2, which shows that our method can still resolve the harder tasks with a success rate of 87% on average. Instead, the three baselines perform much worse, especially for the CoP and RoboGen. CoP fails in all cases because the script policy receives feedback from the environment less frequently and cannot control the speed of the object. For example, the task `throw garlic into bowl` will reset the bowl into some place where the agent cannot reach. However, CoP can only pick and place into a precise position, which fails in the throwing motion. For RoboGen, the reward function has a higher probability of making mistakes without iterative reflection. We also conduct the ablations for the accuracy of the success functions in Sec. 5.3.

### 5.2 POLICY GENERALIZATION ANALYSIS FROM DIVERSE VIDEOS

To further demonstrate the scalability of our framework, we target training a general policy from diverse videos. As mentioned in Sec. 4.3, we regard the Video2Policy as a data engine to generate expert trajectories in simulation. To validate the generalization ability across unseen videos, we focus on one single behavior, lifting up. Specifically, we sample 100 videos concerning the `lifting` behavior from SSv2 dataset, generate the scenes, and train policies for the corresponding task. Afterward, we collect 100 successful trajectories from each policy model, including the $256 \times 256$ size of RGB image observation and the 7-dim actions. Then we train an image-based policy model by imitation learning from the collected trajectories. Finally, we sample another 10 different `lifting` videos and evaluate the performance on those tasks with novel objects.

**Training Details** We choose imitation learning algorithms to learn the general policy and use the Behavior Cloning (BC) implemented by ourselves. For the model architecture, we apply the pre-trained Resnet18 (He et al., 2016) as the backbone to extract features and stack 2 frames as observations.

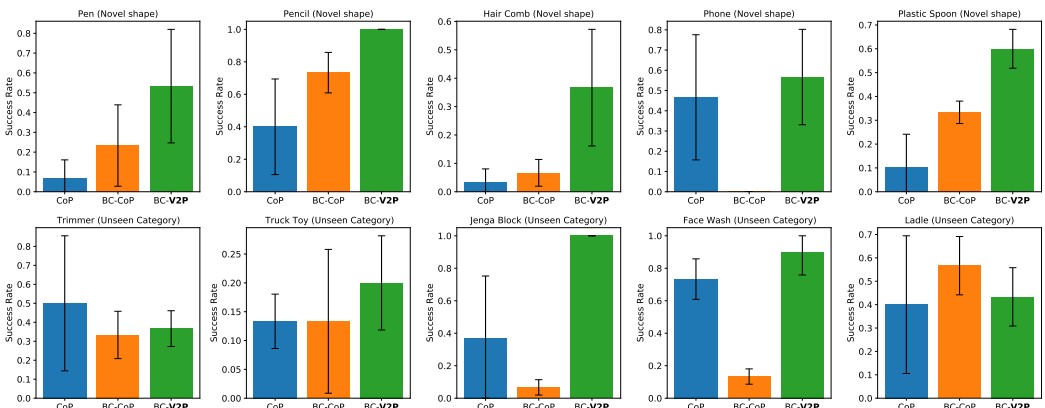

Figure 4: **Performance of the trained general policy on 10 unseen task instances.** BC-V2P outperforms BC-CoP on **9** of 10 significantly and demonstrates stronger generalization ability.

Table 3: **Ablation Results** toward the visual information, success function picking, reward function iteration, and reward function sampling. Removing any component results in a performance drop.

| **Task** (Succ.) | Lifting up | Uncover | Throw | **Avg.** |
|---|---|---|---|---|
| **Video2Policy (V2P)** | $0.93 \pm 0.05$ | $\mathbf{0.97} \pm 0.05$ | $\mathbf{0.70} \pm 0.36$ | $\mathbf{0.87}$ |
| **V2P w.o. visual information** | $0.90 \pm 0.08$ | $0.77 \pm 0.13$ | $0.57 \pm 0.17$ | $0.75$ |
| **V2P w.o. success picking** | $\mathbf{0.97} \pm 0.05$ | $0.43 \pm 0.40$ | $0.30 \pm 0.42$ | $0.57$ |
| **V2P w.o. iterative reward designs** | $0.53 \pm 0.29$ | $0.73 \pm 0.24$ | $0.17 \pm 0.17$ | $0.48$ |
| **V2P w.o. multiple reward samples** | $0.73 \pm 0.12$ | $0.53 \pm 0.24$ | $0.27 \pm 0.38$ | $0.51$ |

Then a policy head is built in a 3-layer MLP with hidden states of 512 dim. Additionally, the policies are trained on the collected trajectories for 30 epochs with a batch size of 1024. The learning rate of the policy head is 3e-4, while set to 3e-5 for the Resnet backbone. For evaluation, we evaluate the final checkpoint on the 10 `lifting` tasks for 3 seeds, and we average the results on 10 trajectories for each seed. The evaluation tasks include 5 objects with novel shapes as well as novel textures, and 5 objects with unseen categories.

**Baseline of General Policy** For general policies, we consider the proposed Video2Policy as a novel data engine for expert trajectory collection. Therefore, other data engines are the baseline, which can generate successful trajectories in our reconstructed scenes. We compare the following models: (1) Code-as-Policy (CoP), which can be applied to the novel tasks directly with state input instead of image observations; (2) BC-CoP, which trains the BC general policy from the data collected by CoP; (3) BC-V2P, same as BC-CoP but using the data collected by V2P. The CoP baseline demonstrates how well the state-based general policy can be, while the latter BC-CoP and BC-V2P results illustrate the performance of the general policy under different data engines.

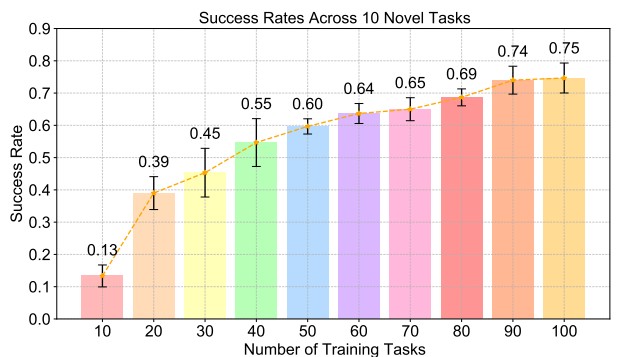

Figure 5: **Scalability of the general policy model (BC-V2P) towards the number of training tasks on `lifting` behavior (3 seeds).** Learning from more videos can enhance the generalization ability of the BC-V2P model.

**Generalization to Unseen Videos** The performances of the models are shown in Fig. 4. After training on 100 task instances, our BC-V2P model works on all the novel tasks and achieves remarkable generalization performance, marked in red. Specifically, the average success rate reaches 75%, while CoP is 32% and the BC-CoP is 26%. It indicates that our proposed Video2Policy framework can obtain more informative policy priors from the videos. Notably, the CoP results are based on

states and the variances from CoP policies are larger than those from V2P policies. And the model performs worse on the objects with unseen categories compared to the objects with novel shapes.

**Scalability Analysis** Furthermore, it is significant to investigate the scalability of our framework towards the number of generated tasks from videos. We analyze it by training the general policy on $N \in \{10, 20, 30, ..., 100\}$ tasks and evaluate the same 10 evaluation tasks, shown in Fig. 5. Overall, increasing the number of videos continues to improve the performance of the BC-V2P model. Note that current real2sim methods only use a few scenes such as 6 scenes (Torne et al., 2024). In contrast, our method can leverage large amounts of scenes and learn policies, which proves the scalability.

### 5.3 ABLATION STUDY

Compared to the previous works on LLM-driven RL (Ma et al., 2023; Wang et al., 2023c), we apply the visual information to propose success functions by LLM and pick the best one by LLM.

**Success Function Evaluation** Moreover, we attempt to calculate the correlation between the generated success functions and the manually designed ones, as shown in Fig. 6. We choose the success rates calculated under different success functions as the data points. Compared to Eureka, the success function generated by more visual information can be more reasonable and closer to the ground-truth ones, with a higher correlation coefficient qual to 0.83. In addition, the generated success functions with visual information can be over-confident, as the data points lie below the y=x curve. Instead, it can be noise with caption only when generating the success functions.

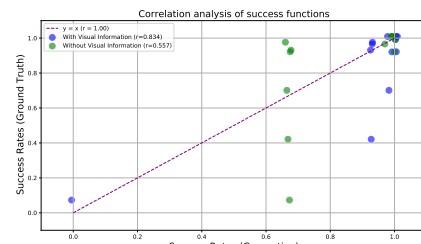

Figure 6: **Correlation analysis** of generated success functions and the manual ground-truth functions.

**Ablation of Code Generation Components** Here, we ablate the results of removing each part in the code generation as follows: (1) V2P w.o. visual information, where only the video caption is provided to generate the codes; (2) V2P w.o. success picking, where we do not pick the best success function by sampling 8 different success functions; (3) V2P w.o. iterative reward designs, where we do not apply the iterative reward reflection to fine-tuning the reward functions; (4) V2P w.o. multiple reward samples, where we only train 1 example of the reward function instead of 8 samples. The results are on the Tab. 3. It will encounter a performance drop without any part. The most significant one can be the iterative reward reflection component, which finetunes the reward function based on the previous training results. Additionally, w.o. the multiple reward samples and w.o. success picking have larger variances than the others. With those techniques, we achieve better performance than the previous LLM-driven methods (Ma et al., 2023; Wang et al., 2023c).

## 6 DISCUSSION

We have proposed Video2Policy, a pipeline for generating simulated tasks from human videos. We show that our design enables us to effectively learn from human videos and generate high quality data. And We show that our data can be used to train a general visuomotor policy that generalizes to unseen tasks. This generalist policy scales favorably with the number of videos used for task generation. We believe this is a step towards generalist robotic policies that can perform a wide range of tasks similar to the wide range of everyday human behavior.

**Limitations and Future work** Our approach is based on existing pre-trained foundation models. As such, it is bottlenecked by the quality of these models, particularly mesh reconstruction and reward code generation. However, as these foundation models continue to improve we expect the performance of our method to improve as well. Another limitation is that it does not fully leverage the dynamic information contained in the video. As for the **future work**, while this paper validates the idea of a pipeline for generating data for generalist robotic policies in simulation, we would like such a pipeline to be useful for real-world policies. Two avenues of future work towards this direction are (i) increasing domain randomization (Andrychowicz et al., 2020) such as by LLM-generated domain randomization schedules (Ma et al., 2024) which will lead to improved sim-to-real transfer, or (ii) improved scene reconstruction as more powerful 3D pretrained models become available (Leroy et al., 2024) which will lead to better visual quality of the produced data.

# 7 REPRODUCIBILITY STATEMENT

The main implementations of our proposed method are in Sec. 4. The settings of the experiments, the details of training policies, and the hyper-parameters are in 5.

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

## A  APPENDIX

### A.1  IMPLEMENTATION DETAILS OF CODE GENERATION

**Examples of Generated Codes** To further illustrate the generated codes described in Sec. 4.2, we take the task `insert fork into storage` as an example, shown in Fig. 7. We provide detailed prompts and examples to inform the GPT-4o for code generation. In the beginning, we generate 8 code samples and then leverage GPT-4o to select the most reasonable one as the base code. Afterward, we perform iterations for the reward function part, during which only the reward function code is required to generate. For each iteration, we learn RL policies under the N different reward functions and select the one with the highest success rates as the base code for the next iteration. Noticed that the VLM can add some states such as `dist_to_fork` and change the observation space automatically.

```
1   ...
2   from factory.tasks.task import Task
3
4   ...
5   scene_info = {'task_class_name': ..., 'objects': ...}
6
7   ...
8   class InsertForkIntoStorageBox(Task):
9       """
10      insert fork into storage box
11      """
12      def __init__(self, envs, urdf_assets_root) -> None:
13          super().__init__(envs, urdf_assets_root, scene_info)
14          self.task_description = "insert fork into storage box"
15      def reset_objects_states(self, env_ids):
16          return super().reset_objects_states(env_ids)
17      def compute_observation_key(self, states):
18          obs_keys = super().compute_observation_key(states)
19          # Add distance observation between the end-effector and the fork
20          dist_to_fork = torch.norm(states["fork_pos"] - states["eef_pos"], dim=-1)
21          states["dist_to_fork"] = dist_to_fork
22          obs_keys.append("dist_to_fork")
23          # Add distance observation between the fork and the storage box
24          dist_fork_to_box = torch.norm(states["fork_pos"] - states["storage_box_pos"], dim=-1)
25          states["dist_fork_to_box"] = dist_fork_to_box
26          obs_keys.append("dist_fork_to_box")
27          return obs_keys
28      def get_observation_space(self):
29          obs_space = super().get_observation_space()
30          # Increase the observation space by the number of new observations added
31          return obs_space + 2
32      def compute_success(self, states):
33          # the fork in the storage
34          # xy position of the fork is within the storage box
35          # z position of the fork is within the storage box
36          xy_distance = torch.norm(states["fork_pos"][:, :2] - states["storage_box_pos"][:, :2], dim=-1)
37          ...
38          # fork should be inside the storage box's boundary in Z-axis
39          z_condition = (lower_z_of_fork < upper_z_of_box) & (upper_z_of_fork > lower_z_of_box)
40          # fork should be inside the storage box's boundary in XY-axis
41          xy_condition = xy_distance < 0.01
42          success = z_condition & xy_condition
43          return success
44      def compute_reward(self, actions, states):
45          return reward_function(actions, states)
46
47  # REWARD FUNCTION
48  @torch.jit.script
49  def reward_function(actions, states):
50      ...
```

Figure 7: An example of the generated code for the task `Insert`.

```
1   ...
2   ### Analysis of Policy Feedback
3   1. **Distance from Gripper to Garlic (dist_grasp_reward)**
4      - **Observation**: The values fluctuate around 0.1, with a minimum of 0.0018 and a maximum of 0.1320.
        ↪  The last value is 0.1272.
5      - **Suggestion**: The fluctuation indicates variability, but the impact may be insufficient for success.
        ↪  Adjust the scale or temperature slightly for better sensitivity.
6   2. **Distance from Garlic to Bowl (dist_garlic_to_bowl)**
7      - **Observation**: The values are quite low (0.0000 - 0.0029), indicating this component is not
        ↪  significantly contributing.
8      - **Suggestion**: Increase the temperature parameter for better gradient.
9   3. **Velocity Reward (vel_reward)**
10     - **Observation**: The values show variability and seem to be contributing to the learning process. The
        ↪  last value is 0.1746.
11     - **Suggestion**: It is effective but might need a slight adjustment in temperature and scaling.
12  4. **Final Position Reward (final_position_reward)**
13     - **Observation**: This component consistently shows zeros, indicating it's not achieved in any of the
        ↪  episodes.
14     - **Suggestion**: Rewriting or significantly adjusting this component is necessary.
15  5. **Total Reward (total_reward)**
16     - **Observation**: The total reward has shown improvement over time but doesn't lead to success. It
        ↪  fluctuates with no episodes achieving success.
17     - **Suggestion**: Improve the component scales to better balance the total reward for successful
        ↪  episodes.
18  ### Key Points for Improvement
19  - Re-scale or rewrite dist_garlic_to_bowl for better contribution.
20  - Rework or introduce final_position_reward to enhance the likelihood of success.
21  - Adjust the scales and temperatures for proper balance.
```

Figure 8: An example of the reward analysis for the task `Throw`.

**Examples of Code Evolutions** Additionally, we also provide an example of modifying the reward function based on the reward logs during training. During the reward iteration stage, we provide examples of how humans adapt and refine the reward functions based on results. Also, we add the logs of each reward component during training as prompts for the next iteration. These logs will be categorized into two groups: from successful trajectories and from failure trajectories. This approach allows the GPT-4o to identify which reward components are most significant in driving success. If the training consistently results in failure without any successful outcomes, we will prompt GPT-4o to redesign the reward function to improve performance and outcomes. The analysis example by the GPT-4o is shown in Fig. 8, and the newly generated reward function example is shown in Fig. 9.

```python
### Improved Reward Function
Based on the analysis and suggestions, here's an improved reward function:
@torch.jit.script
def reward_function(actions, states):
    # type: (Tensor, Dict[str, Tensor]) -> Tuple[Tensor, Dict[str, torch.Tensor]]
    """
    # Chain of Thoughts for reward designs:
    1. Grasp the garlic.
    2. Throw the garlic towards the bowl.
    3. Ensure the garlic reaches the bowl.
    # Reward Components:
    1. The distance between the gripper and the garlic.
    2. The garlic's distance to the bowl.
    3. The velocity of the garlic towards the bowl.
    4. The final position of the garlic being inside the bowl.
    """
    batch_size = states['eef_pos'].shape[0]
    rewards = torch.zeros(batch_size, device=states['eef_pos'].device, dtype=torch.float32)
    reward_components = {}
    # 1. Adjust Distance from gripper to garlic reward
    dist_eef_to_garlic = torch.norm(states['garlic_pos'] - states['eef_pos'], dim=-1)
    dist_grasp_reward = 1 - torch.tanh(8.0 * dist_eef_to_garlic)  # Adjusting TANH temperature
    reward_components['dist_grasp_reward'] = dist_grasp_reward * 0.15  # Updated scale for emphasis
    # 2. Increased temperature for distance between garlic and bowl
    dist_garlic_to_bowl = 1 - torch.tanh(12.0 * states['dist_garlic_to_bowl'])  # Increased temperature for
    ↪    better gradient
    reward_components['dist_garlic_to_bowl'] = dist_garlic_to_bowl * 0.25  # Increased scale to 0.25
    # 3. Velocity reward (slight temperature increase)
    garlic_vel = states['garlic_vel_linear']
    garlic_to_bowl_dir = states['bowl_pos'] - states['garlic_pos']
    garlic_to_bowl_dir = garlic_to_bowl_dir / torch.norm(garlic_to_bowl_dir, dim=-1, keepdim=True)
    vel_towards_bowl = torch.sum(garlic_vel * garlic_to_bowl_dir, dim=-1)
    vel_reward = torch.tanh(4.0 * vel_towards_bowl)  # Slight adjustment in temperature
    reward_components['vel_reward'] = vel_reward * 0.30  # Increased scale to 0.30 for stronger influence
    # 4. Final Position Reward (revised and scaled)
    dist_xy_to_center = torch.norm(states['garlic_pos'][:, :2] - states['bowl_pos'][:, :2], dim=-1)
    upper_z_of_bowl = states["bowl_pos"][:, 2] + states["bowl_size"][:, 2] / 2
    lower_z_of_bowl = states["bowl_pos"][:, 2] - states["bowl_size"][:, 2] / 2
    garlic_in_bowl = ((dist_xy_to_center < (states['bowl_size'][:, 0] / 2)) & (states['garlic_pos'][:, 2] <
    ↪    upper_z_of_bowl) & (states['garlic_pos'][:, 2] > lower_z_of_bowl)).float()
    final_position_reward = garlic_in_bowl * 0.70  # Actually achievable state
    reward_components['final_position_reward'] = final_position_reward
    # Total Reward including penalties
    total_reward = dist_grasp_reward * 0.15 + dist_garlic_to_bowl * 0.25 + vel_reward * 0.30 +
    ↪    final_position_reward * 0.70
    step_penalty = torch.ones_like(total_reward) * 0.01
    rewards = total_reward - step_penalty
    reward_components['total_reward'] = total_reward
    reward_components['step_penalty'] = step_penalty
    return rewards, reward_components
```

Figure 9: An example of the newly generated reward function for the task `Throw`.

## A.2 MORE EXPERIMENTS

**Robustness analysis for each component.** To quantify the robustness of our method, we evaluate vision models used in our pipeline individually by testing their reconstruction success rates. For instance, when assessing the grounding accuracy of DINO, we sample 20 videos. Similarly, for evaluating the segmentation accuracy of SAM-2, we sample 20 successful bounding boxes generated by DINO. Following this approach, we systematically test the robustness of each module in the pipeline. If the reconstruction is broken or identifies the wrong object, it is classified as a failure case. The results are in Tab. 4. We can find that the segmentation and mesh reconstruction parts are more robust than the others.

Moreover, we also conduct experiments to evaluate the noise effects of the depth estimation component. For one thing, we use the depth-aware Realsense Camera to get the ground-truth depth of the

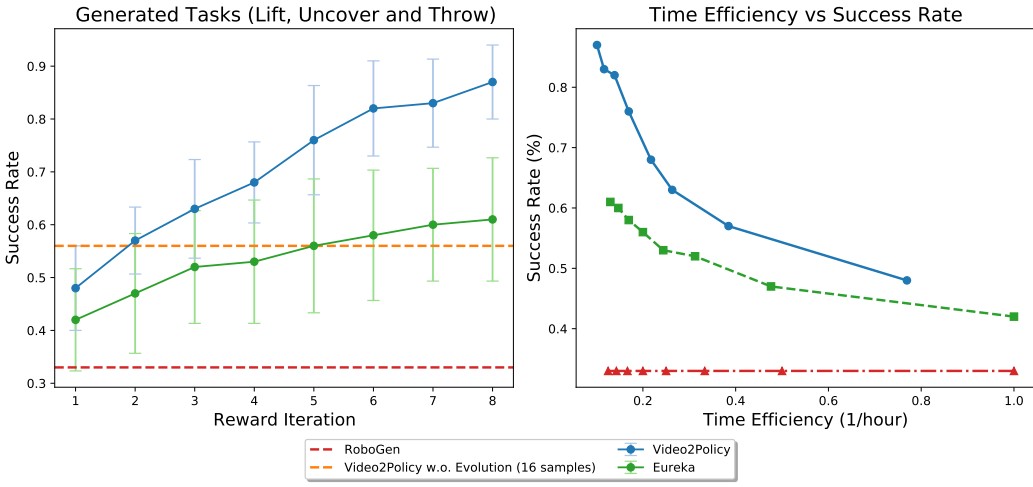

Figure 10: Video2Policy achieves better performance across iteration, which demonstrates better superior Pareto optimality.

Table 4: Failure rates (smaller is better) of the vision models in our pipeline. The average failure rate is 42%, and SAM-2 is the most robust model.

| **Failure Rates** | Grounding DINO | SAM-2 | InstantMesh | FoundationPose | **Avg.** |
| --- | --- | --- | --- | --- | --- |
| SSv2 Videos | 0.60 | **0.15** | 0.35 | 0.55 | 0.42 |

Table 5: D1 distance between the predicted depth by Unidep (Piccinelli et al., 2024) and the ground-truth depth in `Sliding` video.

| | Full size | Center region (0.8x crop) | Object bounding box |
| --- | --- | --- | --- |
| d1 distance | 67.3% | 86.9% | **93.4%** |

Table 6: D1 distance between the predicted depth by Unidep (Piccinelli et al., 2024) and the ground-truth depth in `Sliding` video.

| Object | Remote Control | Mouse |
| --- | --- | --- |
| Predicted Size (l, w, h) | (0.18, 0.04, 0.02) | (0.17, 0.10, 0.05) |
| Ground-truth Size (l, w, h) | (0.21, 0.05, 0.02) | (0.19, 0.08, 0.03) |
| Delta size (l, w, h) | (0.03, 0.01, 0.00) | (0.02, 0.02, 0.02) |

self-recorded video `Sliding` and evaluate the d1 metric (higher is better) for the video (Piccinelli et al., 2024).$\text{d1} = \frac{\text{Number of pixels where } \frac{|d_{\text{pred}} - d_{\text{gt}}|}{d_{\text{gt}}} > 0.1\text{m}}{\text{Total number of pixels}}$. The results are in Tab. 5. The depth estimation of the object region is accurate. And the size error under the depth prediction is as shown in Tab. 6. The error of the size prediction is small. Furthermore, we resize the objects to the GT size and apply the previously trained model to study how the noise affects the final performance. We keep the same state inputs. The performance drops from 97% to 83%, but the model can still solve the task at a high success rate.

**Performance analysis for iterative generation.** Following Eureka (Ma et al., 2023), we visualize the performance of our method and baselines after each evolution iteration in Fig. 10. And we also conduct an ablation study, Video2Policy w.o. Evolution (16 Samples), which only performs the initial reward generation step without iterative improvement. The results are based on the tasks `lifting`, `uncover`, `throw`, which follow the setting in Sec. 5.3. This study examines whether, given a fixed reward function budget, it is more effective to allocate resources toward iterative evolution or simply generate more first-attempt rewards. The results demonstrate that our method signifi-

cantly outperforms the other baselines across multiple iterations. Our method demonstrates superior Pareto optimality, effectively balancing multiple objectives to achieve optimal trade-offs compared to other approaches, shown in Fig. 10.

### A.3 More Ablation Results

All the ablation experiments follow the setting in Sec. 5.3.

**Tracking prompts help for policy learning.** Here we investigate the tolerance of 6D position errors for the tracking part. We conduct an ablation study that removes the 6D position tracking information obtained from FoundationPose in Tab. 7. As shown, after removing the tracking information, the performance drops for the tasks with multiple objects.

Table 7: After removing the tracking information in prompts, the performance drops from 87% to 75%, which is still superior to Eureka.

|  | Lifting Up | Uncover | Throw | Average |
|---|---|---|---|---|
| Video2Policy | $\mathbf{0.93} \pm 0.05$ | $\mathbf{0.97} \pm 0.05$ | $\mathbf{0.70} \pm 0.36$ | $\mathbf{0.87}$ |
| Video2Policy, w.o. tracking info | $0.90 \pm 0.08$ | $0.77 \pm 0.13$ | $0.57 \pm 0.17$ | $0.75$ |
| Eureka | $0.83 \pm 0.13$ | $0.63 \pm 0.38$ | $0.37 \pm 0.29$ | $0.61$ |

```
Example 1:
def compute_observation_key(self, states):
    obs_keys = super().compute_observation_key(states)
    return obs_keys
def get_observation_space(self):
    obs_space = super().get_observation_space()
    return obs_space\
def compute_success(self, states):
    card_lifted = (states['card_pos'][:, 2] - states['table_height']) > (torch.max(states['card_size'])  /
    ↪   2)
    success = card_lifted
    return success

Example 2:
def compute_observation_key(self, states):
    obs_keys = super().compute_observation_key(states)
    dist_gripper_card = torch.norm(states["card_pos"] - states["eef_pos"], dim=-1)
    states["dist_gripper_card"] = dist_gripper_card
    obs_keys.append("dist_gripper_card")
    return obs_keys
def get_observation_space(self):
    obs_space = super().get_observation_space()
    return obs_space + 1
def compute_success(self, states):
    card_height_above_table = states['card_pos'][:, 2] - states['table_height']
    card_lifted = card_height_above_table > 0.2
    success = card_lifted & (states['card_vel_linear'].norm(dim=-1) < 0.01)
    return success
...

Example 2 is more reasonable and better for several reasons:
1. Inclusion of the End-Effector-Card Distance in Observations:
    This is beneficial because it provides an additional critical feature that can help the agent understand
    ↪   its relative position to the target object. This added information is highly useful for learning
    ↪   tasks involving object manipulation.
    ...
2. Enhanced Success Criteria: ...
    The second implementation refines the success criteria by adding a condition that the card's linear
    ↪   velocity should be minimal (states['card_vel_linear'].norm(dim=-1) < 0.01), ensuring that the card
    ↪   is not just lifted but also stable. This is a more precise definition of success for manipulation
    ↪   tasks.
    ...
3. Dynamic and Informative Observations: ...
4. Observation Space Adjustment: ...
```

Figure 11: An example of how GPT-4o picks better code for less hallucination.

**Hallucination issue can be alleviated by picking under GPT-4o.** To evaluate the validity of the generated task code, we have some instructions. For example, for correctness, we inform the GPT-4o to read and analyze the success part; for reasonability, we inform the GPT-4o to avoid picking the code that assumes some scalar value or states. To quantify the frequency of the hallucination,

we run Video2Policy (16 samples, 1 iteration) for the 3 tasks (Lifting, Uncover, Throw). Here we remove the while-loop generation so that not all samples are runnable. (Previously, if one sample fails, we regenerate again until all 8 samples are runnable.) Here the hallucination samples include non-runnable samples and zero-score samples. We can find that after picking by GPT-4o, the hallucination problem alleviates in a degree. Here is an example of how GPT-4o picks better task codes, shown in Fig. 11.

Table 8: Querying GPT-4o for choosing across multiple samples helps alleviate hallucination.

|  | Without Picking | Picking by GPT-4o |
|---|---|---|
| Hallucination | 0.40 | **0.19** |
| Hallucination - Non-runnable | 0.25 | **0.13** |
| Hallucination - Runnable | 0.15 | **0.07** |

**The generated codes do not reset cheating.** Since the task codes are generated under different prompts for all the methods, we conduct ablation studies by choosing the same 'reset' function from our method for the baselines. For the code-as-policy, we generate the policy code and execute it in the Issac Gym. Thus, the 'reset' function is the same as the Video2Policy since they share the task code. We choose 3 tasks, one of a single object and two of multiple objects. The results are illustrated in Tab. 9. For tasks with a single object, the results are the same because the generated 'reset' function calls the base reset function. For tasks with multiple objects, the results have limited changes. The reason is that when generating the reset function, the LLM introduces certain constants. These constants may vary. However, the variance has limited effects on the final results. It indicates that there is no reset cheating.

Table 9: Using the same 'reset' function as the Video2Policy, the baselines have limited changes for the evaluation results. This proves that the better results of our method do not come from 'reset' cheating.

|  | Lifting Up | Uncover | Throw | Average |
|---|---|---|---|---|
| Video2Policy | **0.93** ± 0.05 | **0.97** ± 0.05 | **0.70** ± 0.36 | **0.87** |
| Code-as-Policy | 0.33 ± 0.21 | 0.10 ± 0.08 | 0.00 ± 0.00 | 0.14 |
| RoboGen | 0.28 ± 0.09 | 0.67 ± 0.26 | 0.03 ± 0.05 | 0.33 |
| RoboGen (same reset function) | 0.28 ± 0.09 | 0.60 ± 0.28 | 0.03 ± 0.05 | 0.30 |
| Eureka | 0.83 ± 0.13 | 0.63 ± 0.38 | 0.37 ± 0.29 | 0.61 |
| Eureka (same reset function) | 0.83 ± 0.13 | 0.67 ± 0.21 | 0.37 ± 0.29 | 0.62 |

### A.4 EXPERIMENTS OF SIM2REAL

Although this work aims to leverage internet videos for simulation policy learning, we also conduct some sim2real experiments to verify the effectiveness. Specifically, following Sec. 5.2, we collect 200 trajectories from each reconstructed scene in simulation for 100 `lifting` tasks. Then we train a general policy via imitation learning and subsequently deployed the policy in the real-world setting.

To alleviate the sim-to-real gap, we employed the following two strategies:

**Input Representation and Network Architecture.** We take as input the 0/1 segmentation masks of the image, the robot's end-effector (EEF) state, and the gripper state. SAM-2 is adopted for segmentation, where the pixel position of the target object is provided manually as input in the first frame, shown in Fig. 12. We stack 2 frames and add an additional multi-layer perceptron (MLP) layer to map the robot state into a 256-dimensional feature vector. Furthermore, the rotation component of the action is scaled by a factor of 0.2 for better stability.

**Domain Randomization.** During data collection in the simulation, randomization is applied to the actions with noise levels of 0.02 and a random delay of 0.01–0.02 seconds. Moreover, the physical

Figure 12: Examples of the 0-1 Segmentation Mask Observation between the simulation and the real, which can better bridge the sim2real gap.

properties of the object, such as size and weight, are also randomized. We ensure consistency between the camera poses in simulation and the real world.

In real-world experiments, the object's position varies within a 10 cm range. The image input had a resolution of 256x256. In terms of the setup, we use Franka robotic arms, Robotiq grippers, and Stereolabs cameras. We evaluate the performance of the policy towards lifting a mouse, a cup, and a piece of bread. Notably, while there are some mouse and bottle objects in the simulation, the bread is absent in the collected simulation dataset and is soft.

Here the general `lifting` policy achieves a success rate of 72% across 10 novel objects in simulation. The sim2real results are as shown in Tab. 10. Compared to the 72% success rate in simulation, it achieves 47% success rate in the real world. It proves the efficacy of our pipeline, which builds the pipeline of internet videos to policies. We notice that gripping slippery objects, such as a cup, pose challenges for the robot, resulting in a relatively low success rate. For the bread, the task was relatively easier despite the object being unseen in the simulation. This can be attributed to the segmentation mask observation and the bread's relatively large surface area, which facilitates successful manipulation.

Overall, these experiments demonstrate that the general policy trained in simulation possesses effective sim-to-real transfer capabilities. Additionally, the results highlight the potential of the proposed Video2Policy pipeline, underscoring its effectiveness in enabling good performance, scalability, and deployment in real-world scenarios.

|  | Mouse | Cup | Bread | Average |
|---|---|---|---|---|
| Succ. | 0.50 | 0.40 | 0.50 | 0.47 |

Table 10: The success rates of the learned policy for `lifting` tasks on real robots.

