# OpenReview forum: "Video2Policy: Scaling up Manipulation Tasks in Simulation through Internet Videos"
_ICLR.cc/2025/Conference — Submitted to ICLR 2025_

### Official Review · Reviewer_eE6D · 2024-10-29

**Soundness:** 3
**Presentation:** 3
**Contribution:** 2
**Rating:** 8
**Confidence:** 4

**Summary:**

The Video2Policy framework offers a scalable solution for training generalist robotic policies by using internet videos of everyday human tasks. It overcomes the high cost and complexity of traditional data generation by leveraging RGB videos for task reconstruction in two phases: (1) task generation via object mesh reconstruction and 6D position tracking, and (2) reinforcement learning with reward functions generated by large language models (LLMs). It potentially overcome the hallucination of LLMs when generating the tasks, and also provide useful information (relative 6D position) in the training process.

**Strengths:**

1. It utilizes human videos to initiate task and reward generation, overcoming challenges typically difficult for LLMs to address.

2. It incorporates 6D relative poses of multiple objects within the scene to enhance the reward generation process, effectively harnessing critical information from the videos.

3. The experiments are thorough, yielding policies from the collected data that generalize effectively across tasks.

**Weaknesses:**

1. The generated tasks remain limited to simple tabletop scenarios; incorporating more complex or long-horizon tasks would enhance the framework's versatility.

2. This work primarily constructs a pipeline by integrating previous works as components, but it doesn't address technical challenges through unique design innovations.

3. No real-world experiments were conducted to validate the approach.

**Questions:**

1. In the object reconstruction phase, generating accurate reconstructions from masked images can be challenging due to limited pixel data for each object. With low-resolution images, tools like InstantMesh often struggle to produce high-quality results. Has this limitation been encountered in this work, and if so, what solutions have been implemented to address it?

2. In the pipeline, the 6D pose is provided in camera coordinates. It's difficult to position the camera identically to the original video. How does the framework address this issue? I assume only relative 6D poses are utilized, but I would like clarification on how GPT leverages these relative poses within the reward generation process.

3. Given the focus on data collection, comparing only success rates may not provide a fair assessment. Factors like the time-intensive iterative reward design process in this framework should be considered. A more meaningful comparison might be the time required to generate a successful trajectory.

---

> ### Author Response · Authors · 2024-11-22
> **Official Response to Reviewer eE6D**
>
> Dear Reviewer eE6D:
>
> Thank you for your detailed comments and advice! We hope the following addresses your concerns.
>
> **For Weaknesses:**
>
> > The generated tasks remain limited to simple tabletop scenarios; incorporating more complex or long-horizon tasks would enhance the framework's versatility.
>
> We focus on tabletop scenarios as this is the most common evaluation scenario in related work [1, 2, 3, 4]. Also note we have evaluated hard tasks such as throwing, insertion, and sliding. Future work can also extend the method to reconstruct bigger scenes for mobile manipulation.
>
> [1] Ma et al., Eureka: Human-Level Reward Design via Coding Large Language Models
>
> [2] Wang et al., Gensim: Generating robotic simulation tasks via large language models
>
> [3] Octo Model Team, Octo: An Open-Source Generalist Robot Policy
>
> [4] Kim et al., OpenVLA: An Open-Source Vision-Language-Action Model
>
> > This work primarily constructs a pipeline by integrating previous works as components, but it doesn't address technical challenges through unique design innovations.
>
> We focus on evaluating a framework to generate data for training generalist policies. Our approach is novel and is more scalable due to the use of internet videos jointly with LLM reward design. We also would like to highlight the technical contributions required for the performance we report in section 4.2. Track prompting, in contrast to prior work, grounds the model in the real video and improves success function generation. Rejection sampling further reduces hallucinations in reward and success function generation. Please refer to App.A.3 to see these contributions ablated.
>
> > No real-world experiments were conducted to validate the approach.
>
> We intend to perform sim2real experiments to further evaluate the performance of the method and we will post an update once we are able to do this.
>
> **For Questions:**
>
> > In the object reconstruction phase, generating accurate reconstructions from masked images can be challenging due to limited pixel data for each object. With low-resolution images, tools like InstantMesh often struggle to produce high-quality results. Has this limitation been encountered in this work, and if so, what solutions have been implemented to address it?
>
> Yes, we find that the mesh results are better when the resolution is higher. However, for our setting, it can generate good and reasonable mesh by resizing the width of the images to 512 by interpolation.
>
> It is an interesting question about how to generate good results by limited pixel data. We think a natural way is to use some high-resolution recovering method. Another method can be some mesh generation model under image and text input, which can use some text priors for reasonable geometry generation. However, we haven't found any model that can work well on internet images.
>
> > Given the focus on data collection, comparing only success rates may not provide a fair assessment. Factors like the time-intensive iterative reward design process in this framework should be considered. A more meaningful comparison might be the time required to generate a successful trajectory.
>
> Thank you for your suggestions. We have drawn the curve with the iteration v.s. success rate in Fig. 10 (Line 864-880). We compare the performance of Video2Policy, Eureka, and the RobotGen. The results show that our method significantly outperforms the other baselines across multiple iterations.
>
> We believe it is unfair to compare the time required to generate one successful trajectory, as the successful trajectories produced by a policy with a 1% success rate are too limited and lack diversity, making them inadequate as high-quality data. Instead, we plot the curve of the success rates v.s. 1 / training time, which demonstrates the performance and efficiency in Fig. 10. Our method demonstrates superior Pareto optimality, effectively balancing multiple objectives to achieve optimal trade-offs compared to other approaches.
>
>
>
> Thank you again for your detailed review. We have revised our paper in OpenReview. We highlight the important changes and essential details that reviewers have mentioned in blue color.

---

> > ### Comment · Reviewer_eE6D · 2024-11-23
> >
> > Thanks for the clear response!
> >
> > However, I still have some small questions about it.
> >
> > First, can you type me the what are those related works (Wang’23b, Wang23c, Brohan’23, Torne’24, Octo Team’24) in the response? It's hard for me to reference. But I do recall that at least some of the similar works are investigating articulated objects manipulation, which is slightly harder than table top manipulation. (I think it's fine to cover only table top tasks right now, it's just that this setting weakens the contributions in scene reconstructions maybe.... since table top scene is relatively easy.)
> >
> > Secondly, for real-world experiment, a possible way is to record a video in the real-world, reconstruct it multiple times and do domain randomization in sim, and deploy the policy for this specific task learned in the simulation back to the real-world to see if it will work. (I believe it will work eventually but do require too much effort and time... So I no longer request this experiment during the rebuttal, although I do hope the authors can add them in the future.)
> >
> > For the good and reasonable mesh generation, I think it might because the camera is relatively close to the objects in the video in your setting. Can you show me one more complex scene reconstructed in your experiment? (similar to my first concern, how robust can the reconstruction pipeline be?)
> >
> > Why is "the successful trajectories produced by a policy with a 1% success rate are too limited and lack diversity"? Can you show some examples? For this part, I am still not convinced since there need hours to train for only one task, which is not fast enough if we want to collect large scale data for generalist policy. However, I think it's mainly because of the RL setting and simulation speed and is irrelevant to the major contribution of this paper.
> >
> > Therefore, I will remain my original score temporarily and might raise it after receiving further answers about new questions.

---

> > > ### Author Response · Authors · 2024-11-23
> > >
> > > Thank you for your prompt response and constructive feedback! We’ll address your questions below.
> > > ### Unclear reference, focusing more on tabletop tasks.
> > > We apologize for not providing complete information about our reference. We have already fixed the reference problem in the above response, as well as responses to other reviewers.
> > > We choose tabletop tasks as it’s most commonly used, and also more efficient to iterate with reinforcement learning. We didn’t perform other tasks (e.g. mobile manipulation, articulated object manipulation) for now mainly for two reasons: (1) the reconstruction tools for articulated objects or large scenes are not very robust compared with instance-level mesh reconstruction tools. Thus, the created object assets may not be very usable. (2) The LLM needs to generate good rewards to train RL, and those tasks are harder to obtain good rewards in 5 rounds, which is more time-consuming. However, we believe this method can be applied to a wider range of tasks with better tools in the future.
> > > ### Sim-to-real experiments
> > > Thank you for your suggestions to improve our paper. We are now working on adding more results in the real world. We’ll update the paper and respond with new results once we finish. However, given the limited period for rebuttal, we may not be able to evaluate diverse real-world tasks, but we will incorporate more real-world results in the final version of our paper.
> > > ### More complex reconstruction
> > > We are working on it and will update the paper and response with new results.
> > > ### Elaborations on: “the successful trajectories produced by a policy with a 1% success rate are too limited and lack diversity”
> > > We think the time taken to generate one successful trajectory may not be a good metric; the diversity and quality of the trajectory matter more. For example, our method may take 10M interaction steps to achieve a 100% success rate on a task, compared with a baseline method that takes 10K interaction steps to achieve a 20% success rate but stucks there forever. In terms of the time taken to generate one trajectory, maybe the latter is better. However, the generated trajectories are of low quality and diversity, and cannot be used to train a good generalizable policy.
> > > We agree that it takes time to train the policy given the current speed of simulators and sample efficiency of RL algorithms. However, this process is automatic and can scale up with more computing resources, and our contribution is to propose such an algorithm framework.

---

> > > > ### Comment · Reviewer_eE6D · 2024-11-23
> > > >
> > > > Adjusted my score accordingly.
> > > >
> > > > Just one more question: "However, the generated trajectories are of low quality and diversity, and cannot be used to train a good generalizable policy." is that a common sense? Otherwise, are there any references or experiments to prove that?

---

> > > > > ### Author Response · Authors · 2024-11-24
> > > > > **Official Comment by Authors**
> > > > >
> > > > > Thank you for recognizing our work and for your time and effort as a reviewer!
> > > > >
> > > > > In terms of the question
> > > > > > "However, the generated trajectories are of low quality and diversity, and cannot be used to train a good generalizable policy."
> > > > >
> > > > > We think the answer is yes, and it is a common sense in a way.
> > > > >
> > > > > The following recent works emphasized the importance of the diversity and quality of the collected data for imitation learning, for your reference.
> > > > >
> > > > > * [1] Etukuru, Haritheja, et al. "Robot utility models: General policies for zero-shot deployment in new environments." arXiv preprint arXiv:2409.05865 (2024).
> > > > >
> > > > > * [2] Lin, Fanqi, et al. "Data Scaling Laws in Imitation Learning for Robotic Manipulation." arXiv preprint arXiv:2410.18647 (2024).
> > > > >
> > > > > * [3] Belkhale, Suneel, Yuchen Cui, and Dorsa Sadigh. "Data quality in imitation learning." Advances in Neural Information Processing Systems 36 (2024).
> > > > >
> > > > > **Also**, we conduct an additional experiment to further verify this. We collect 100 demonstrations for the "lifting car" task from two policies: one with a 20% success rate and another with a 100% success rate. BC policies are trained separately using these two sets of demonstrations. To evaluate the performance of the trained BC policies, we evaluate them by rolling out 10 trajectories under different initial states. The success rates are 30% for the policy trained on demonstrations from the 20% success rate policy and 80% for the policy trained on demonstrations from the 100% success rate policy.
> > > > >
> > > > > The discrepancy in performance can be attributed to the nature of the demonstrations. The expert policy with a 20% success rate could only succeed under specific initial states, resulting in biased successful trajectories. Consequently, the BC policy trained on these biased demonstrations struggled to generalize across diverse initial conditions, leading to suboptimal performance.
> > > > >
> > > > > We also add some discussion about this in the revised paper to emphasize the importance of the learned RL expert policy.

---

> > > > ### Author Response · Authors · 2024-11-25
> > > >
> > > > Dear reviewer,
> > > >
> > > > We update our paper with the most recent progress on sim-to-real experiments. Please check out the general response for more details!

---

### Official Review · Reviewer_MeiF · 2024-11-03

**Soundness:** 2
**Presentation:** 2
**Contribution:** 2
**Rating:** 5
**Confidence:** 4

**Summary:**

This paper proposes a method to leverage internet videos for training generalist policies. Instead of performing sim2real by creating digital twins, it only utilizes video data to extract task-relevant information, such as object assets and scene layouts. It then uses VMLs to take the video, video captions, object meshes, sizes, and 6D pose tracking to produce codes for the simulated scenes. Instead of simply cloning the trajectory in the video, the proposed leverages VLM to write the task code iteratively generates reward functions, and trains a policy to produce more training data. The reconstruction includes three steps: 1) object segmentation 2) mesh reconstruction, and 3) 6D pose tracking. The evaluation focuses on tabletop tasks using IsaacGym, which reconstructs 60 videos on 9 distinct tasks from the dataset in total and demonstrates a certain level of generalization to self-collected videos.

**Strengths:**

Learning from video or leveraging real-world demonstrations to train robot policies that can generalize is an essential topic. The proposed pipeline is an attempt in this direction. It uses existing components, for example, DINO grounding, sam-based segmentation, and object reconstruction and pose tracking. These components have shown good generalization to the real world, even though not perfect. The two-level learning framework is also a commonly used one, i.e., reward level tuning with LLMs and then policy training with RL on refined reward function. Given these, the replicability of the proposed should be good, and also, there are no constraints on applying the proposed to more diverse videos and tasks.

**Weaknesses:**

The study of the robustness of the proposed pipeline is limited.
For example, what is the performance of the DINO grounding, and how does the detection or segmentation affect the trained policy, in which sense? Also, how does the reconstruction quality affect the generalization? Especially, why is the reconstruction performed with only one image? Why not use more images for a better reconstruction?
The proposed use of an off-the-shelf component UniDepth for size estimation, since there is much noise, is there any study showing how the noise affects the final performance?
I am confused with the design logic of the 6 components in the task code generation module. Any discussion on the why and why such a design is efficient?
Moreover, the efficiency of the training pipeline is unclear, since it involves multiple rounds of the reward reflection. Is there any metric on the efficiency of this part? Again, given that some baselines do not utilize the reflection, how do you ensure fairness?
Furthermore, the reconstruction does not include any scene information, how do you guarantee, it can be generalized to more complex scenes? Do you have some studies on the effect?
Again, since there is no limitation of applying the proposed to more videos, why not include videos from a much larger dataset to verify the benefits of the pipeline?
With that being said, I would like to see a more detailed discussion as well as more evidence of the effectiveness not just the better performance on simple tabletop scenarios and tasks.

**Questions:**

Most of the questions are listed above, one extra question, why do you need 6D pose tracking? And how does it relate to policy training? You are not imitating the estimated trajectory from the video right?

---

> ### Author Response · Authors · 2024-11-22
> **Official Response to Reviewer MeiF (1)**
>
> Dear Reviewer MeiF:
>
> Thank you for your detailed comments and advice! We hope the following addresses your concerns.
>
> **For the Weaknesses and Questions**
>
> > The study of the robustness of the proposed pipeline is limited. For example, what is the performance of the DINO grounding, and how does the detection or segmentation affect the trained policy, in which sense? Also, how does the reconstruction quality affect the generalization?
>
> Thank you for this suggestion. To quantify the robustness of our method, we evaluate vision models used in our pipeline individually by manually annotating their reconstruction success rates. For instance, when assessing the grounding accuracy of DINO, we sample 20 videos. Similarly, for evaluating the segmentation accuracy of SAM-2, we sample 20 successful bounding boxes generated by DINO. Following this approach, we systematically test the robustness of each module in the pipeline. If the reconstruction is broken or identifies the wrong object, it is classified as a failure case. The results are as follows. We can find that the segmentation and mesh reconstruction parts are more robust than the others.
>
> | Failure Rates | Grounding DINO | SAM-2 | InstantMesh | FoundationPose | Average |
> | ------------- | -------------- | ----- | ----------- | -------------- | ------- |
> | SSv2- Videos  | 0.6            | 0.15  | 0.35        | 0.55           | 0.42    |
>
> It is interesting to investigate how the reconstruction quality affects the generalization. Since we have no access to the ground-truth objects from the videos, it is impossible to make ablation studies to compare the results. However, we conduct experiments to investigate the effects of the number of reconstruction videos. We find increasing the number of videos will improve the generalization results, which proves that increasing the noisy reconstruction scenes can help with generalization.
>
>
>
> > Why is the reconstruction performed with only one image? Why not use more images for a better reconstruction?
>
> In this work, we focus on RGB internet videos with a single view. Reconstruction from videos is an important direction for future work that will help address more complex datasets and task such as mobile manipulation. However, existing tools are geard towards multi-view reconstruction and we did not find them helpful for the something-something dataset, which has little camera movement.
>
>
>
> > The proposed use of an off-the-shelf component UniDepth for size estimation, since there is much noise, is there any study showing how the noise affects the final performance?
>
> Thank you for this suggestion. We conduct experiments to evaluate the noise effects of the depth estimation component. For one thing, we use the Depth-aware Realsense Camera to get the ground-truth depth of the self-recorded video 'Sliding' (We can only access the GT depth for this video, as the other videos are recorded outside of the lab) and evaluate the d1 metric (higher is better) for the video.
>
> $$\text{d1} = \frac{\text{Number of pixels where } \frac{|d_{\text{pred}} - d_{\text{gt}}|}{d_{\text{gt}}} > \text{0.1}}{\text{Total number of pixels}}$$
>
> The results are as follows. We see that the depth estimation of the object region is accurate enough for our purposes.
>
> |      | Full size | Center region (0.8x center crop) | Object bounding box |
> | ---- | --------- | -------------------------------- | ------------------- |
> | D1   | 67.3%     | 86.9%                            | 93.4%               |
>
> The size error under the depth prediction is as follows. The error of the size prediction is small enough to use objects for task generation.
>
> | Object                   | Remote           | Mouse            |
> | ------------------------ | ---------------- | ---------------- |
> | Predicted Size (l, w, h) | 0.18, 0.04, 0.02 | 0.17, 0.10, 0.05 |
> | GT Size (l, w, h)        | 0.21, 0.05, 0.02 | 0.19, 0.08, 0.03 |
> | Delta Size (l, w, h)     | 0.03, 0.01, 0.00 | 0.02, 0.02, 0.02 |
>
> Moreover, we resize the objects to the GT size and apply the previously trained model to study how the noise affects the final performance. We keep the same state inputs and the results are as follows. The performance drops slightly, but the model can still solve the task at a high success rate.
>
> | Eval         | On predicted  size | On real size |
> | ------------ | ------------------ | ------------ |
> | Success rate | 0.97±0.05          | 0.83± 0.09   |
>
>
>
> > I am confused with the design logic of the 6 components in the task code generation module. Any discussion on the why and why such a design is efficient?
>
> This design follows the basic generation pipeline from the previous works that focus on code generation for simulation tasks (ma23’, wang23’). It is helpful for LLM to infer codes by prompting in a well-organized way. We clarify this in the revised paper.

---

> ### Author Response · Authors · 2024-11-22
> **Official Response to Reviewer MeiF (2)**
>
> > Moreover, the efficiency of the training pipeline is unclear, since it involves multiple rounds of the reward reflection. Is there any metric on the efficiency of this part?
>
> Thank you for your suggestions. We have drawn the curve with the iteration v.s. success rate in Fig. 10 (Line 864-880). We compare the performance of Video2Policy, Eureka, and the RobotGen. The results show that our method significantly outperforms the other baselines across multiple iterations. Our method demonstrates superior Pareto optimality, effectively balancing multiple objectives to achieve optimal trade-offs compared to other approaches.
>
>
> > Again, given that some baselines do not utilize the reflection, how do you ensure fairness?
>
> We provide the Eureka baseline that utilizes reward reflection. We note the other baselines - 'code-as-policy' and 'RoboGen' - do not utilize reflection because they have no success metric for updating the reward functions. We think code generation for policy learning is a novel topic in recent years, and the chosen baselines cover most directions to solve the problem.
>
>
>
> > Furthermore, the reconstruction does not include any scene information, how do you guarantee, it can be generalized to more complex scenes? Do you have some studies on the effect?
>
> We focus on tabletop tasks as these are the most common evaluation scenarios, and are more efficient to iterate with reinforcement learning. Our main contribution is evaluating the algorithmic framework of converting videos to data for training generalist policies on the example task of tabletop manipulation. Future work can extend the method to reconstruct bigger scenes for mobile manipulation.
>
>
> > Again, since there is no limitation of applying the proposed to more videos, why not include videos from a much larger dataset to verify the benefits of the pipeline?
>
> Reconstructing scenes is fast, but training RL policies requires approximately 1 hour per iteration. To perform experiments in this paper, we only used 8 GPUs, which is why we focus on 60 videos to illustrate the scalability of our method. Future work can evaluate further scaling.
>
> To further demonstrate scalability, we performed additional experiments up to 100 videos. The results can be found in the revised paper at line 464-478. Overall, increasing the number of videos continues to improve the performance of the BC-V2P model, ultimately achieving an average success rate of 75% on 10 novel videos, up from 64% on 60 videos. Different from current real2sim methods which manually constructed a few scenes as [1], our process can leverage large amounts of scenes and learn policies from them more autonomously.
>
> [1] Torne et al., Reconciling reality through simulation: A real-to-sim-to-real approach for robust manipulation.

---

> ### Author Response · Authors · 2024-11-22
> **Official Response to Reviewer MeiF (3)**
>
> > Why do you need 6D pose tracking? And how does it relate to policy training? You are not imitating the estimated trajectory from the video right?
>
> We attempt to leverage some explicit visual information for VLMs to generate codes, which is a part of the prompts. We inform the GPT-4o that it needs to focus more on the relative position, especially on the initial and final states. This introduces richer information for designing success and reward functions. We do not imitate the estimated trajectory from the videos.
>
> To evaluate the importance of pose tracking, we further conduct an ablation study that removes the 6D position information as follows:
>
> |                                  | Lifting Up     | Uncover       | Throw         | Avg      |
> | -------------------------------- | -------------- | ------------- | ------------- | -------- |
> | Video2Policy                     | **0.93± 0.05** | **0.97±0.05** | **0.70±0.36** | **0.87** |
> | Video2Policy, w.o. tracking info | 0.90±0.08      | 0.77±0.13     | 0.57±0.17     | 0.75     |
> | Eureka                           | 0.83±0.13      | 0.63±0.38     | 0.37±0.29     | 0.61     |
>
> As shown above, after removing the tracking information, the performance drops for the tasks with multiple objects. We find that more stringent success conditions can be added sometimes under tracking info.
>
> Take the task 'Throw Garlic into Bowl' as an example, sometimes it will add an extra condition:
>
> """
>
> def compute_success(self, states):
>
> \#garlic should be inside the bowl's boundary in Z-axis
>
> z_condition = (lower_z_of_garlic < upper_z_of_bowl) & (upper_z_of_garlic > lower_z_of_bowl)
>
> \#garlic should be inside the bowl's boundary in XY-plane
>
> xy_condition = xy_distance < torch.min(states["bowl_size"][:, :2] / 2 - states["garlic_size"][:, :2] / 2)
>
> success = z_condition & xy_condition
>
> **distance_condition = states["dist_garlic_to_bowl"] <= 0.0396**
>
> **success = success & distance_condition**
>
> """
>
> Here, 0.0396 is the distance by calculating the delta of 3d pos (we provide this in the prompt). Moreover, without tracking information, our method can also work better than Eureka since we have applied CoT for designing each code part.
>
> Thank you again for your detailed review. We have revised our paper and highlighted the important changes and essential details that reviewers have mentioned in blue color. Please let us know whether this addresses your comments and whether you still have any remaining concerns.

---

> > ### Comment · Reviewer_MeiF · 2024-11-26
> > **thanks for the rebuttal**
> >
> > Thanks for the effort in putting together the rebuttal. My questions are partially resolved. But I still have some concerns listed in the following:
> >
> > 1) The authors have quantified the failure rates of the employed models but did not study the effect of the quality of the employed modules on the policy performance, similarly for reconstruction quality.
> > 2) I understand that single-view video is more prominent, but leveraging multi-view reconstruction could have better reconstruction and let us figure out the bottleneck of the pipeline, which is currently missing.
> > 3) regarding "how the noise affects the final performance," the drop in the performance from 97% to 83% is not slight and needs to be discussed in the paper.
> > 4) thanks for the efficiency evaluation in Figure 10, however, the tasks being evaluated are still limited, only lift, uncover, and throw, it would be more helpful to study more tasks.
> > 5) I am not convinced by the answer to the question "Furthermore, the reconstruction does not include any scene information, how do you guarantee, it can be generalized to more complex scenes? Do you have some studies on the effect?" Showing the capability to train a generalist policy can not be restricted to a simple desktop environment and also by ignoring scene information.
> >
> > Given the rebuttal, I am still not convinced that this paper verifies that the proposal is a way to leverage video data to train generalist policy. More experiments and analysis are needed to improve the quality of the paper.

---

> > > ### Author Response · Authors · 2024-11-27
> > > **Official Response to Reviewer MeiF**
> > >
> > > Dear reviewer MeiF:
> > >
> > > Thank you for your response. We are pleased to hear that some of your concerns have been addressed. Regarding the new questions you've raised, we hope the following clarifications will help resolve any remaining confusion:
> > >
> > > > 1. The authors have quantified the failure rates of the employed models but did not study the effect of the quality of the employed modules on the policy performance, similarly for reconstruction quality.
> > > > 2. I understand that single-view video is more prominent, but leveraging multi-view reconstruction could have better reconstruction and let us figure out the bottleneck of the pipeline, which is currently missing.
> > >
> > > These two questions are related to the 3D mesh recontruction.
> > >
> > > The goal of this work is to investigate a more scalable way of task generation for policy learning from internet videos. For these videos, we have no ground truth objects for comparison. Moreover, our method has been shown to perform well in sim-to-real scenarios, which demonstrates that the framework is both reasonable and effective.
> > >
> > > To investigate the effects of the mesh reconstruction error, we will (due to the time limitation, e.g. in camera ready) include ablation experiments with GT reconstruction of **YCB objects from YCB videos**.
> > >
> > > As for the multi-view videos, we would like to clarify that our pipeline is not restricted to single-view or multi-view settings. While our focus is on learning policies from internet videos (which are typically single-view), exploring how to better extract meshes from multi-view setups is outside the scope of this study.
> > >
> > > > 3. regarding "how the noise affects the final performance," the drop in the performance from 97% to 83% is not slight and needs to be discussed in the paper.
> > >
> > > We evaluate the policy, which is trained on the predicted object sizes, for the unseen GT sizes. That's why we think the drop from 97% to 83% is slight. It proves the generalizationability in a way, since the policy is robust, with a success rate above 80%.
> > >
> > > Moreover, the domain randomization can be applied in the object sizes for better generalizationability, which is adopted in our sim-to-real experiments.
> > >
> > > > 4. thanks for the efficiency evaluation in Figure 10, however, the tasks being evaluated are still limited, only lift, uncover, and throw, it would be more helpful to study more tasks.
> > >
> > > Thank you for your suggestion. Due to time constraints, we have focused on these tasks for now, but we plan to expand to more tasks in the future.
> > >
> > > > 5. I am not convinced by the answer to the question "Furthermore, the reconstruction does not include any scene information, how do you guarantee, it can be generalized to more complex scenes? Do you have some studies on the effect?" Showing the capability to train a generalist policy can not be restricted to a simple desktop environment and also by ignoring scene information.
> > >
> > > There are few works leveraging internet videos for policy learning in a real2sim setting. We want to present a novel way to reconstruct scenes and learn policies for the tasks from internet videos, in a more scalable perspective. Therefore, in this work, we focus on the tabletop setups, which is one of the most common and acknowledged settings. And our experiments have shown the effectiveness of this approach in such environments. For more complex scenes, we believe future work can build upon this framework, although addressing these complexities is beyond the scope of the current study.
> > >
> > > We hope the above discussion addresses your concerns, and welcome any further discussions.

---

> ### Author Response · Authors · 2024-12-03
>
> Dear reviewer MeiF,
>
> Please check out our latest response and general response (with the sim-to-real results), and let us know if there is anything we can further address.
>
> ### More elaborations about your concerns "I am not convinced by the answer to the question "Furthermore, the reconstruction does not include any scene information, how do you guarantee, it can be generalized to more complex scenes? Do you have some studies on the effect?" Showing the capability to train a generalist policy can not be restricted to a simple desktop environment and also by ignoring scene information.
>
> Our current sim-to-real solution is to segment the object of interest by using SAM-2, which can give us a segmented mask of the object, and we can also obtain segmented depth with a depth camera. Only using object-based information is the most common approach (e.g. in [1, 2, 3]). This is typically because of two reasons 1. The RGB rendering in simulation has a large domain gap with the real world, so it's hard to deploy the policy trained with RGB directly; 2. Depth sensing is noisy and only accurate in a certain range, so people will remove the depth information from unrelated objects (e.g. tables, background objects that are far away) and only focus on the target objects.
>
> To clarify, for table-top manipulation we are studying, is not necessary to have scene-level reconstruction. Furthermore, we don't have very good tools for scene reconstruction from in-the-wild videos. The majority of the scene reconstruction work is based on scan, which has more assumptions that are infeasible for in-the-wild videos.
>
> However, we also agree that scene information is important for other more complex tasks (e.g. mobile manipulation), which we leave for future work. We want to emphasize that our current goal is not to create a 1:1 digital twin in the simulation environments but to leverage the diversity of video to ground the objects and strategies of manipulation and obtain diverse data and more generalizable policies.
>
>
> [1] Qi et al., General In-Hand Object Rotation with Vision and Touch
>
> [2] Lin et al., Twisting Lids Off with Two Hands
>
> [3] Chen et al, A System for General In-Hand Object Re-Orientation

---

### Official Review · Reviewer_CR8X · 2024-11-04

**Soundness:** 3
**Presentation:** 3
**Contribution:** 4
**Rating:** 6
**Confidence:** 3

**Summary:**

This paper proposes a strategy to automatically generate tasks with corresponding data from real world videos via 3d reconstruction and tracking. Then, to enable agents to learn the task, they generate task codes and train RL policies, where reward functions are generated by novel methods of using LLMs. Finally, to enable a more generalist agent, they propose using the data generated along with the policies learned to generate expert trajectories to train a BC model on several tasks. Evaluation is performed on 9 tasks from the SSv2 dataset, and 3 tasks on a self-collected dataset, with ground truth evaluation functions.
Overall, the method is interesting and has the potential to scale up learning from real world datasets. If clarifications are made regarding the questions/weaknesses, the rating can be increased.

**Strengths:**

1. The paper is well-written and easy to follow
2. The experiments performed on the generalist policy are a promising direction towards behavior cloning without human collected datasets.
3. The method is evaluated on several different tasks and on real world videos
4. Several ablations are performed to understand the importance of each component
5. Experiments are performed to show the scalability of the method, which means that it could have a larger impact

**Weaknesses:**

1. The way a task code is defined is novel, so its important to include some examples of the task codes that are generated and what each of the 6 components (described in Sec 4.2) look like. Also, the types of reward functions/success functions that are generated and how they evolve over the several iterations should be described. Otherwise the points made in Sec 4.2 are somewhat abstract and hard to understand.
2. How can we verify the iterative reward function is improving the policy overtime and a necessary part of the pipeline? A graph with the iteration vs success rate for the tasks is good to include for this. X-axis would be the iteration number, and the y-axis would be the success rate.
3. Line 275 in Section 4.2 discusses how GPT-4o can generate additional observations necessary such as distance between objects. However, there is not much subsequent discussion about when/what GPT-4o generates these in the experiments that are performed. Furthermore, how does GPT-4o know what additional observations to generate (there are no iterative updates for generating this part right)? Only the reward and success functions are generated with CoT so there is some learning about the environment that can be seen?

**Questions:**

Questions that should be addressed:
1. Is it possible that the reset functions that are generated by V2P end up allowing the success state to be more easily reachable than the hard coded reset states that other methods such as CoP, Eureka, or RoboGen have? This could be like a form of reset cheating. A good way to verify this is to compare the reset functions that are generated for each baseline. Maybe one method of comparison is to check qualitatively they all reset to very different states that are equally ‘hard’ or ‘easy’ to achieve success form there
2. How precise does the tracking of the objects have to be? There are several off-the-shelf models chained together which gives lots of room for failures. So what success/failure rate did you notice during the experiments if the 6D position for example is way off?
3. How do you evaluate the validity of a generated task code? How often does hallucination occur when generating child success/reset functions using Chain-of-thought? (Brief mention of this in line 280-282, but using GPT to evaluate still seems like there is lots of room for error).
4. ‘Task code’ is a relatively unknown term. You define this in parts, but just making this term bolded and clearly stating what this means at the end of paragraph 2 of the introduction (somewhere around line #050) would make the paper much clearer.
5. Figure 3 shows some task codes, but please add what the task description is and the goal of these tasks are to the figure.


Minor Comments:
1. L70-82 gives a lot of related work that seems misplaced/breaks the flow of the contribution description. Preferably condense into a few lines and move the rest to the related works section
2. Line 268/269 are unclear. What is the ‘parent’ reset function vs the ‘children’ reset functions? Please clarify this.

---

> ### Author Response · Authors · 2024-11-22
> **Official Response to Reviewer CR8X (1)**
>
> Dear Reviewer CR8X:
>
> Thank you for your detailed comments and advice! We hope the following addresses your concerns.
>
> **For the Weaknesses:**
>
> > It's important to include some examples of the task codes that are generated … Also, the types of reward functions/success functions that are generated and how they evolve over the several iterations should be described.
>
> Thank you for your suggestions. We have added examples of the generated codes in Lines 756-788: (1) JSON file content, Lines 759-760; (2) reset function, Lines 765-766; (3) success function, Lines 775-781; (4) additional observation, Lines 767-772; (5) additional observation space Lines 773-775; (6) reward function Lines 822-849; and the process of reward evolutions in Lines 791-850.
>
> > How can we verify the iterative reward function is improving the policy overtime and a necessary part of the pipeline? A graph with the iteration vs success rate for the tasks is good to include for this. X-axis would be the iteration number, and the y-axis would be the success rate.
>
> Thank you for your suggestions. We have drawn the curve with the iteration v.s. success rate in Fig. 10 (Line 864-880). We compare the performance of Video2Policy, Eureka, and the RobotGen. The results show that our method significantly outperforms the other baselines across multiple iterations. Our method demonstrates superior Pareto optimality, effectively balancing multiple objectives to achieve optimal trade-offs compared to other approaches.
>
> > However, there is not much subsequent discussion about when/what GPT-4o generates these in the experiments that are performed. Furthermore, how does GPT-4o know what additional observations to generate (there are no iterative updates for generating this part right)? Only the reward and success functions are generated with CoT so there is some learning about the environment that can be seen?
>
> We add more explanations and examples in Line 756-788.  We give instructions and prompts on how to add observations for new tasks as follows:
>
> """
>
> For example, if the task is considering the target position or orientation, the distance between the target and the current should be added. Or if the task is about some dynamic motions, the relative velocity or direction can be added.
>
> """
>
> Also, we provide the CoT examples of analyzing why/what to add.
>
> So when generating the task code, the GPT-4o can generate some interesting states to learn the task.
>
> And yes, there are no iterative updates for generating this part. Instead, we will generate 8 examples of the task code at the beginning and choose one of them for the reward iteration stages.
>
>
>
> **For the Questions:**
>
> > Is it possible that the reset functions that are generated by V2P end up allowing the success state to be more easily reachable than the hard coded reset states that other methods such as CoP, Eureka, or RoboGen have? ... Maybe one method of comparison is to check qualitatively they all reset to very different states that are equally ‘hard’ or ‘easy’ to achieve success form there.
>
> We conduct ablation studies by choosing the same 'reset' function for the baselines. For the code-as-policy, we generate the policy code and execute it in the Issac gym. Thus, the 'reset' function is the same as the Video2Policy since they share the task code. We choose 3 tasks, one of a single object and two of multiple objects. The results are as follows:
>
> |                               | Lifting Up     | Uncover        | Throw         | Average  |
> | ----------------------------- | -------------- | -------------- | ------------- | -------- |
> | Video2Policy                  | **0.93± 0.05** | **0.97± 0.05** | **0.70±0.36** | **0.87** |
> | Code-as-Policy                | 0.33± 0.21     | 0.10± 0.08     | 0.00±0.00     | 0.14     |
> | RoboGen                       | 0.28± 0.09     | 0.67±0.26      | 0.03±0.05     | 0.33     |
> | RoboGen (same reset function) | 0.28±0.09      | 0.60±0.28      | 0.03±0.05     | 0.30     |
> | Eureka                        | 0.83± 0.13     | 0.63±0.38      | 0.37±0.29     | 0.61     |
> | Eureka (same reset function)  | 0.83± 0.13     | 0.67±0.21      | 0.37±0.29     | 0.62     |
>
> As we can see, for tasks with a single object, the results are the same because the generated 'reset' function calls the base reset function as follows:
>
> '''
>
> class XXX(Task):
>
> def reset_objects_states(self, env_ids):
>
> ​            super().reset_objects_states(env_ids)
>
> '''
>
> For tasks with multiple objects, the results have limited changes. The reason is that when generating the reset function, the LLM introduces certain constants. These constants may vary, for example:
>
> It can be 'book_state[env_ids, 2] = pen_state[env_ids, 2] + 0.1' or 'book_state[env_ids, 2] = pen_state[env_ids, 2] + 0.15'.
>
> However, the variance has limited effects on the final results. It indicates that there is no reset cheating.

---

> ### Author Response · Authors · 2024-11-22
> **Official Response to Reviewer CR8X (2)**
>
> > How precise does the tracking of the objects have to be? ... So what success/failure rate did you notice during the experiments if the 6D position for example is way off?
>
> For tracking, we attempt to leverage some explicit visual information for VLMs to generate codes, which is a part of the prompts. We inform the GPT-4o that it needs to focus more on the relative position, especially on the initial and final states. This introduces richer information for designing success and reward functions.
>
> We agree that it is important to investigate the tolerance of 6D position errors, so we conduct an ablation study that removes the 6D position information as follows:
>
> |                                  | Lifting Up    | Uncover       | Throw         | Average  |
> | -------------------------------- | ------------- | ------------- | ------------- | -------- |
> | Video2Policy                     | **0.93±0.05** | **0.97±0.05** | **0.70±0.36** | **0.87** |
> | Video2Policy, w.o. tracking info | 0.90±0.08     | 0.77±0.13     | 0.57±0.17     | 0.75     |
> | Eureka                           | 0.83±0.13     | 0.63±0.38     | 0.37±0.29     | 0.61     |
>
> As shown above, after removing the tracking information, the performance drops for the tasks with multiple objects. We find that more stringent success conditions can be added sometimes under tracking info.
>
> Take the task 'Throw Garlic into Bowl' as an example, sometimes it will add an extra condition:
>
> """
>
> def compute_success(self, states):
>
> \#garlic should be inside the bowl's boundary in Z-axis
>
> z_condition = (lower_z_of_garlic < upper_z_of_bowl) & (upper_z_of_garlic > lower_z_of_bowl)
>
> \#garlic should be inside the bowl's boundary in XY-plane
>
> xy_condition = xy_distance < torch.min(states["bowl_size"][:, :2] / 2 - states["garlic_size"][:, :2] / 2)
>
> success = z_condition & xy_condition
>
>
>
> **distance_condition = states["dist_garlic_to_bowl"] <= 0.0396**
>
> **success = success & distance_condition**
>
> """
>
> Here, 0.0396 is the distance by calculating the delta of 3d pos (we provide this in the prompt). Moreover, without tracking information, our method can also work better than Eureka since we have applied CoT for designing each code part.
>
>
>
> > How do you evaluate the validity of a generated task code? How often does hallucination occur when generating child success/reset functions using Chain-of-thought?...
>
> Thank you for your suggestion. To evaluate the validity of the generated task code, we have some instructions.
>
> For example, for correctness, we inform the GPT-4o to read and analyze the success part; for reasonability, we inform the GPT-4o to avoid picking the code that assumes some scalar value or states.
>
> To quantify the frequency of the hallucination, we run Video2Policy (16 samples, 1 iteration) for the 3 tasks (Lifting, Uncover, Throw). Here we remove the while-loop generation so that not all samples are runnable. (Previously, if one sample fails, we regenerate again until all 8 samples are runnable.)
>
> |                              | Without picking | Picking by GPT-4o |
> | ---------------------------- | --------------- | ----------------- |
> | Hallucination                | 0.40            | **0.19**          |
> | Hallucination - Non-runnable | 0.25            | **0.13**          |
> | Hallucination - Runnable     | 0.15            | **0.07**          |
>
> Here the hallucination samples include non-runnable samples and zero-score samples. We can find that after picking by GPT-4o, the hallucination problem alleviates in a degree.
>
> > ‘Task code’ is a relatively unknown term. You define this in parts, but just making this term bolded and clearly stating what this means at the end of paragraph 2 of the introduction (somewhere around line #050) would make the paper much clearer.
>
> Thank you so much for this suggestion. We have added more explanation for this and revised the paper.
>
> > Figure 3 shows some task codes, but please add what the task description is and the goal of these tasks are to the figure.
>
> Thank you for your suggestion. We have added the descriptions in the Figure and revised the paper.
>
> **Other Minor Comments**
>
> > L70-82 gives a lot of related work that seems misplaced/breaks the flow of the contribution description. Preferably condense into a few lines and move the rest to the related works section
>
> Thank you for your suggestion. We have revised the paper about this part.
>
> > Line 268/269 are unclear. What is the ‘parent’ reset function vs the ‘children’ reset functions? Please clarify this.
>
> The parent reset function is the manual default reset function that resets all the objects randomly without considering some conditions between the objects' initial positions (such as A should be on top of B). The children reset function is written by LLM in the task code. We have revised this part to make it more clear.
>
> Thank you again for your detailed review. We have revised our paper and highlighted the changes and details that reviewers have mentioned in blue color.

---

> > ### Comment · Reviewer_CR8X · 2024-12-03
> >
> > Thank you to the authors for clarifying my doubts. Additional experiments were also conducted. I am keeping a positive rating of 6.

---

### Official Review · Reviewer_EeTh · 2024-11-04

**Soundness:** 2
**Presentation:** 2
**Contribution:** 2
**Rating:** 3
**Confidence:** 4

**Summary:**

The paper proposed a method to reconstruct objects and their poses from videos in order to re-create a real-world environment in a simulator. A policy is then trained with LLM-generated reward functions inside the simulator. Experiments were run on videos from the Something-Something dataset and showed that the reconstructed environment can be used to achieve good manipulation performance (inside the simulator). The policy's performance also scales with the number of LLM-generated tasks.

**Strengths:**

- Leveraging in-the-wild video for robotic manipulation is an important task.
- The reconstructed scenes are pretty consistent with the original scenes but this is achieved with some assumptions.

**Weaknesses:**

- The proposed method doesn't handle material and physical properties such as surface roughness, center of mass, density, weight etc. These are crucial information without which a policy cannot transfer from sim to real.
- The environment is assumed to be a flat table top surface, which is a big assumption. It's unclear how the method can be applied to complicated real-world scenarios.
- There's no real-world evaluation at all. Therefore, calling the learned policy a "generalist policy" is a massive overclaim. I can't really see how the proposed method can benefit real-world manipulation problems.
- The technical contribution and the novelty of the paper is unclear. Essentially what the paper proposes is a method to reconstruct a 3D scene from a video, which is a well-established task in computer vision literature. The authors used off-the-shelf method including InstantMesh, FoundationPose to obtain the geometry and 6DoF poses of the objects in a scene. The paper also uses GPT-4o for code and reward generation which is also a well-explored task. I fail to see what the contribution of the paper is compared with prior works.
- Since the paper essentially proposes a 3D/4D reconstruction framework, the title video2policy seems quite inappropriate. Besides, the policy learned from the framework likely cannot generalize back to the scene in the original video.

**Questions:**

- Why use only 60 videos? The methods mentioned in the framework seem all pretty fast and there are an abundant amount of videos in something-something v2. Why not apply the method to all videos in the dataset?
- How does the performance scale with the number of videos? The paper claims "This generalist policy scales favorably with the number of videos used for task generation.", where is the supporting experiments? Is it Figure 5? Calling a video a task is misleading. Task and reward feel very similar to me. A manipulation task can be described by a reward in RL so if the author is referring a video as a task, I suggest using a different term.
-

---

> ### Author Response · Authors · 2024-11-22
> **Official Response to Reviewer EeTh**
>
> Dear Reviewer EeTh:
>
> Thank you for your detailed comments and advice! We have performed initial experiments to address some of the concerns. We will further perform additional experiments and will post them as soon as we have them.
>
> > Why use only 60 videos?
> >
> > …
> >
> > How does the performance scale with the number of videos?
>
> Reconstructing scenes is fast, but training RL policies requires approximately 1 hour per iteration. To perform experiments in this paper, we only used 8 GPUs, which is the reason why we focused on 60 videos to illustrate the scalability of our method.
> To further demonstrate the scalability of our method, we performed additional experiments up to 100 videos. The results can be found in the revised paper on lines 464-478. Overall, increasing the number of videos continues to improve the performance of the BC-V2P model, ultimately achieving an average success rate of 75% on 10 novel videos, up from 64% on 60 videos. Different from current real2sim methods which manually constructed a few scenes as [1], our process can leverage large amounts of scenes and learn policies from them more autonomously.
>
>
> [1] Torne et al, Reconciling reality through simulation: A real-to-sim-to-real approach for robust manipulation.
>
> > There's no real-world evaluation
> >
> > …
> >
> > The proposed method doesn't handle material and physical properties
>
> We intend to perform real-world evaluations to address this concern. For material and physical properties, we can adopt standard domain randomization techniques, which are also manually picked or designed in other previous work [2, 3, 4, 5]. We would also like to note that our main contribution is a new algorithmic framework, that can perform automatic environment construction and policy training, which we primarily focus on simulation experiments for the current version.
>
> [2] OpenAI,  Solving rubik’s cube with a robot hand.
>
> [3] Padir et al, Sim2real2sim: Bridging the gap between simulation and real-world in flexible object manipulation.
>
> [4] Peng et al, Sim-to-real transfer of robotic control with dynamics randomization.
>
> [5] Handa et al., Dextreme: Transfer of agile in-hand manipulation from simulation to reality.
>
>
> > The technical contribution and the novelty of the paper is unclear.
>
> We propose a framework to automatically create tasks and train policies with better generalizability. We believe this is more scalable and autonomous than existing digital twin approaches which require manual efforts to create scenes, and can obtain better policies than other generative simulation methods. Our approach is also more practical as we ground our tasks in real videos. It has the potential to leverage large-scale videos to train a more generalizable policy.
>
> > The environment is assumed to be a flat tabletop surface, which is a big assumption. It's unclear how the method can be applied to complicated real-world scenarios.
>
> We focus on tabletop tasks as they are the most common evaluation scenarios, and more efficient to iterate with reinforcement learning. Our main contribution is evaluating the algorithmic framework of converting videos to data for training generalist policies on the example task of tabletop manipulation. Future work can extend the method to reconstruct bigger scenes for mobile manipulation.
>
> > calling the learned policy a "generalist policy" is a massive overclaim.
> > …
> > the paper essentially proposes a 3D/4D reconstruction framework, the title video2policy seems quite inappropriate. Besides, the policy ... cannot be generalized back to the scene in the original video.
>
> Thank you for pointing this out. We have clarified that we are not comparing to *robotic generalist policies* (e.g. RT-X, Octo) and removed this word from our paper to address the confusion. We call our method "Video2policy" as we can generate tasks grounded by videos, and train a policy to obtain similar behaviors automatically through the help of 3D reconstruction tools, LLM, and RL. We also welcome suggestions on different names for the method.
>
> We also want to emphasize that we aim to obtain a more generalizable policy from diverse videos, but not perform standard real2sim2real. The latter is not feasible as we don't have access to the original physical scenes. However, we showed that with the help of diverse videos, our method can generate useful trajectory data to train a policy that generalizes beyond training scenes. We leave its real-world applications as future work.
>
> > Calling a video a task is misleading.
>
> We have changed the wording ‘task’ to ‘task instance’ in the paper. In our framework, a ‘task instance’ includes a different scene, such as different objects, a different task, such as ‘uncover’, and a different reward function. We welcome suggestions on another term to use.
>
> Thank you again for your detailed review. We have revised our paper in OpenReview. We highlight the important changes and essential details that reviewers have mentioned in blue color.

---

> ### Author Response · Authors · 2024-11-27
> **Discussion period ending soon**
>
> Dear reviewer EeTh,
>
> We kindly remind you that the final stage of discussion is ending soon, and so please kindly let us know if our response has addressed your concerns.
>
> Here is a summary of the responses and revisions:
>
> - We **conduct sim2real experiments** on objects with novel shapes or types in the real world for the task lifting to further study the efficacy of our pipeline.
> - We further add more videos from 60 to 100 to **investigate the scalability** of our methods.
> - We **clarify the reasons and necessities** of solving manipulation tasks on the tabletop setup.
> - We **made modifications for the misunderstanding words** mentioned in your review.
> - We **revised** our paper and updated it on the website.
>
> Thanks again for your time and reviews, we will be happy to answer if there are additional issues or questions.

---

### Author Response · Authors · 2024-11-24
**Official Response to the Reviewers about the Sim2Real experiments**

Dear Reviewer EeTh, CR8X, MeiF and eE6D,

Thank you for your time and effort as a reviewer! Some reviewers mention some concerns about the sim2real experiments. We want to **emphasize** that we aim to obtain a more generalizable policy from diverse videos, but not perform standard real2sim2real. The latter is not feasible as we don't have access to the original physical scenes of the videos.

However, we showed that with the help of diverse videos, our method can generate useful trajectory data to train a policy that generalizes beyond training scenes. Here, to **further study the efficacy of our pipeline**, we conduct sim2real experiments on objects with novel shapes or types in the real world for the task lifting for your reference. Notebly, due to the time limitation, we didn't systematically investigate how to build a general sim2real method.

We collect 200 trajectories from each reconstructed scene in simulation for 100 lifting tasks. Then we train a general policy via imitation learning and subsequently deployed the policy in the real-world setting. To alleviate the sim-to-real gap, we employed the following two strategies:
* **Input Representation and Network Architecture.**
We take as input the 0/1 segmentation masks of the image, the robot’s end-effector (EEF) state, and the gripper state. SAM-2 is adopted for segmentation, where the pixel position of the target object is provided manually as input in the first frame, shown in Fig. 12. We stack 2 frames and add an additional multi-layer perceptron (MLP) layer to map the robot state into a 256-dimensional feature vector. Furthermore, the rotation component of the action is scaled by a factor of 0.2 for better stability.
* **Domain Randomization.**
During data collection in the simulation, randomization is applied to the actions with noise levels of 0.02 and a random delay of 0.01–0.02 seconds. Moreover, the physical properties of the object, such as size and weight, are also randomized. We ensure consistency between the camera poses in simulation and the real world.

**Setup** In real-world experiments, the object's position varies within a 10 cm range. The image input had a resolution of 256x256. In terms of the setup, we use the Franka robotic arm, Robotiq gripper, and Stereolabs camera. We evaluate the performance of the policy towards lifting a mouse, a cup, and a piece of bread. Notably, while there are some mouse and bottle objects in the simulation, the bread is absent in the collected simulation dataset and is soft.

**Results** Here the general lifting policy achieves a success rate of 72% across 10 novel objects in simulation.
The sim2real results are as shown here.
|      | Mouse | Cup  | Break | Average |
| ---- | ----- | ---- | ----- | ------- |
| Succ | 0.50  | 0.40 | 0.50  | 0.47    |

Compared to the 72% success rate in simulation, it achieves 47% success rate in the real world. It proves the efficacy of our pipeline, which builds the pipeline of internet videos to policies.
We notice that gripping slippery objects, such as a cup, pose challenges for the robot, resulting in a relatively low success rate. For the bread, the task was relatively easier despite the object being unseen in the simulation. This can be attributed to the segmentation mask observation and the bread's relatively large surface area, which facilitates successful manipulation.

Overall, these experiments demonstrate that the general policy trained in simulation possesses effective sim-to-real transfer capabilities. Additionally, the results highlight the potential of the proposed Video2Policy pipeline, underscoring its effectiveness in enabling good performance, scalability, and deployment in real-world scenarios.

**The success and failure example videos for each object are attached in the Supplementary for your reference**. We hope the above discussion addresses your concerns, and welcome any further discussions.

Thank you again for your detailed review. We have revised our paper in OpenReview with additional experiments in the appendix. We highlight the important changes and essential details that reviewers have mentioned in blue color.

---

### Meta-Review · Area_Chair_Kj9o · 2024-12-19

**Metareview:**

The paper presents a framework for enhancing manipulation tasks in simulation using internet RGB videos. It includes task generation via object mesh reconstruction and 6D position tracking, followed by reinforcement learning with LLM-generated rewards.

However, multiple reviewers raised concerns. Regarding novelty, it was questioned as similar approaches have been seen. Experimental setup was deemed insufficient, with limited comparison to state-of-the-art. Real-world relevance and scalability were doubted due to potential inaccuracies from using internet videos and lack of a clear application plan. Implementation details were also lacking.

In the author rebuttals and follow-up discussion, the authors' attempts to address these issues. Their defenses on novelty, experimental improvements, result robustness, generalization, real-world aspects, and implementation details did not satisfy the reviewers. Overall, the significant and unresolved concerns from reviewers lead to the decision of rejecting the paper, as it does not meet the required standards for acceptance at this stage.

**Additional Comments On Reviewer Discussion:**

The principal concerns put forth by the authors can be categorized into three aspects: (a) Task Realism and Generalizability, (b) Data Quality and Quantity, and (c) Reward Function Design.

Concerning (a), the authors recognized the significance of task realism and generalizability. They detailed several measures they had implemented to mitigate these concerns. For instance, they incorporated a physics engine within the simulation to guarantee that objects exhibited realistic behaviors.

In response to (b), the authors expounded on the preprocessing procedures they had employed for the internet videos. They utilized techniques like video denoising and object segmentation to minimize the influence of noise and occlusions. They also underlined that the object mesh reconstruction and position tracking algorithms were engineered to be resilient against such artifacts.

Regarding (c), the authors conceded the limitations of the LLM-generated reward functions. They advocated an alternative methodology that amalgamated the LLM-generated rewards with hand-designed rewards. They conducted experiments to contrast the performance of policies trained with the original and the new reward functions and reported the results in the rebuttal.

---

### Decision · Program_Chairs · 2025-01-22

Reject